# Neural circuit-wide analysis of changes to gene expression during deafening-induced birdsong destabilization

**Bradley M Colquitt[1,2]\*[†], Kelly Li[1,2], Foad Green[1,2‡], Robert Veline[1,2§], Michael S Brainard[1,2]\***

[1]Howard Hughes Medical Institute, Chevy Chase, United States; [2]Department of Physiology, University of California, San Francisco, San Francisco, United States

**\*For correspondence:** colquitt@ucsc.edu (BMC); michael.brainard@ucsf.edu (MSB)

**Present address:** [†]Department of Molecular, Cell, and Developmental Biology, University of California-Santa Cruz, Santa Cruz, United States; [‡]Syapse, Inc, San Francisco, United States; [§]The Advanced Science Research Center, The City University of New York, The Graduate Center at the City University of New York, New York, United States

**Abstract** Sensory feedback is required for the stable execution of learned motor skills, and its loss can severely disrupt motor performance. The neural mechanisms that mediate sensorimotor stability have been extensively studied at systems and physiological levels, yet relatively little is known about how disruptions to sensory input alter the molecular properties of associated motor systems. Songbird courtship song, a model for skilled behavior, is a learned and highly structured vocalization that is destabilized following deafening. Here, we sought to determine how the loss of auditory feedback modifies gene expression and its coordination across the birdsong senso-rimotor circuit. To facilitate this system-wide analysis of transcriptional responses, we developed a gene expression profiling approach that enables the construction of hundreds of spatially-defined RNA-sequencing libraries. Using this method, we found that deafening preferentially alters gene expression across birdsong neural circuitry relative to surrounding areas, particularly in premotor and striatal regions. Genes with altered expression are associated with synaptic transmission, neuronal spines, and neuromodulation and show a bias toward expression in glutamatergic neurons and *Pvalb/Sst*-class GABAergic interneurons. We also found that connected song regions exhibit correlations in gene expression that were reduced in deafened birds relative to hearing birds, suggesting that song destabilization alters the inter-region coordination of transcriptional states. Finally, lesioning LMAN, a forebrain afferent of RA required for deafening-induced song plasticity, had the largest effect on groups of genes that were also most affected by deafening. Combined, this integrated transcriptomics analysis demonstrates that the loss of peripheral sensory input drives a distributed gene expression response throughout associated sensorimotor neural circuitry and identifies specific candidate molecular and cellular mechanisms that support the stability and plasticity of learned motor skills.

## Editor's evaluation

This is an important study that uses the song system in a bird model to understand the transcriptional mechanisms underlying neuronal adaptations to sensory deprivation. The manuscript offers compelling data in support of the authors' hypothesis that these transcriptional changes are related to song plasticity. The work will be of interest to biologists who study neuronal plasticity mechanisms.

## Introduction

The accurate and stable performance of motor skills relies on sensory feedback (*Todorov, 2004*). The loss of this feedback, for example through hearing or vision loss from injury or neurodegeneration, can

lead to increased errors in the execution of even well-learned motor behaviors, such as speech and walking (*Lane and Webster, 1991*; *Waldstein, 1990*; *Wood et al., 2011*). Yet it is poorly understood how such peripheral sensory loss influences the properties of central motor circuits to drive neural plasticity and how these effects in turn influence motor output.

The courtship song of songbirds, a learned motor skill subserved by a dedicated and discrete neural architecture, offers a tractable system in which to characterize the neural mechanisms that underlie sensorimotor integration and motor skill stability. Juvenile birds produce unstructured and variable vocalizations that, over the course of several months of learning, become more structured, less variable, and more similar to adult song (*Brainard and Doupe, 2013*). In finches, after this developmental learning period, birdsong performance remains extraordinarily consistent from rendition-to-rendition over the course of a bird's life and is said to be 'crystallized.' However, auditory feedback plays an essential role in maintaining this stability; modifying auditory feedback or completely removing auditory input through deafening can drive changes to birdsong (*Brainard and Doupe, 2001*; *Brainard and Doupe, 2000*; *Fukushima and Margoliash, 2015*; *Leonardo and Konishi, 1999*; *Lombardino and Nottebohm, 2000*; *Nordeen and Nordeen, 1992*; *Okanoya and Yamaguchi, 1997*; *Tschida and Mooney, 2012*; *Woolley and Rubel, 1997*). The neural mechanisms that underlie these changes have been studied in terms of physiology and morphology, yet we lack a transcriptome- and circuit-wide understanding of how altered auditory feedback influences gene expression in song sensorimotor circuitry — a critical biological vantage point to understand how sensory information intersects with the nervous system to influence motor plasticity.

To gain insight into the molecular and cellular factors that regulate the stability of adult birdsong, we analyzed gene expression alterations in birdsong sensorimotor circuitry and surrounding non-song regions in response to deafening, a strong driver of song destabilization. We developed a gene expression profiling approach that enabled large-scale analysis of gene expression in spatially defined brain regions. Using this technique, we identified a suite of expression changes across the song system, including region-specific and common transcriptional responses as well as altered gene expression correlations across regions. Using a previously generated single-cell atlas of the premotor portion of song neural circuitry, we identified the cellular types that experience the greatest transcriptional change following deafening. Finally, we examined how input from a song region required for deafening-induced song plasticity influences gene expression in its song premotor target and found a diverse set of expression changes, with substantial overlap with those elicited by deafening.

## Results
### Deafening destabilizes birdsong and increases song variability
Songbirds rely on auditory feedback to maintain the quality of their songs (*Figure 1A*). Past work has shown that experimentally removing this feedback by deafening results in the gradual deterioration of both song spectral structure and temporal ordering of the individual elements (syllables) that comprise song (*Nordeen and Nordeen, 1992*; *Okanoya and Yamaguchi, 1997*; *Woolley and Rubel, 1997*). Deafening also drives a range of physiological, cellular, and molecular changes in the song system, including alterations to neuronal turnover (*Scott et al., 2000*; *Wang et al., 1999*) (but see *Pytte et al., 2012*), dendritic spine morphology (*Peng et al., 2012a*; *Peng et al., 2013*; *Tschida and Mooney, 2012*; *Zhou et al., 2017*), neuronal excitability (*Tschida and Mooney, 2012*), and gene expression (*Watanabe et al., 2002*).

We reasoned that comparisons of gene expression in birds undergoing song destabilization following deafening would uncover molecular pathways involved in either promoting or limiting song plasticity. We, therefore, generated a cohort of eighteen adult male Bengalese finches (*Lonchura striata domestica*) that were either deafened through bilateral cochlear removal or underwent a sham surgery (nine birds for each condition, *Figure 1B*). This cohort was further divided into sets of birds that were euthanized 4, 9, or 14 days post-procedure (three birds per procedure type and time point). This range of time points was used to generate a diversity of song destabilization values for subsequent gene expression analysis. Birds were euthanized two hours after lights-on. As in previous studies (*Brainard and Doupe, 2000*; *Okanoya and Yamaguchi, 1997*; *Tschida and Mooney, 2012*; *Woolley and Rubel, 1997*), deafening caused a gradual decay of song quality over the course of several days, while sham surgery induced relatively little song change (*Figure 1B–H*, *Figure 1—figure supplement*

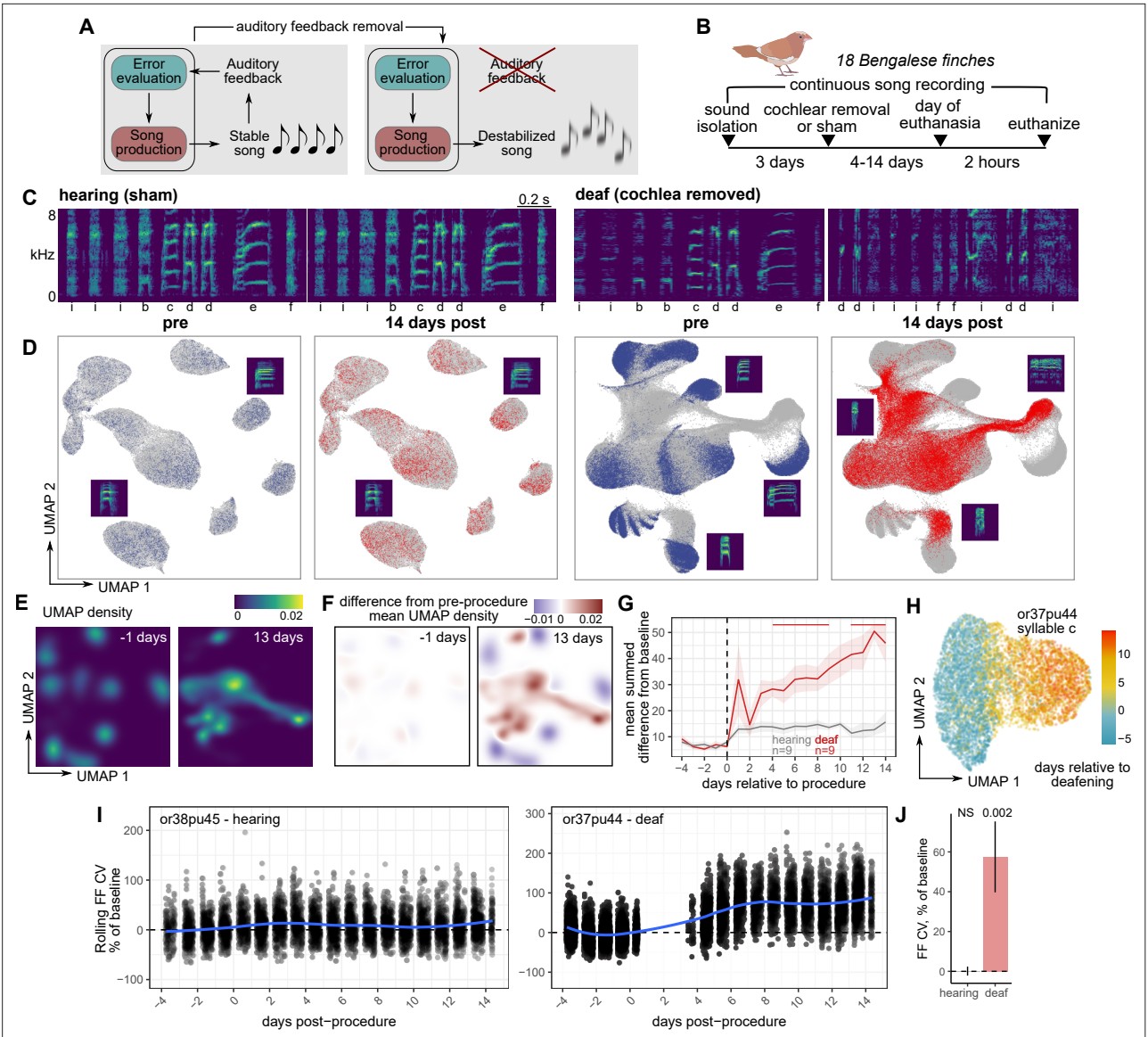

**Figure 1.** Rapid and global destabilization of the song following deafening. (**A**) Song destabilization through the removal of auditory feedback. Adult songbirds use auditory feedback to evaluate their own song production and maintain song quality and consistency. Loss of auditory feedback results in the gradual destabilization of song. (**B**) Experimental overview. After a baseline period of song recording, Bengalese finches (*Lonchura striata domestica*) were either deafened through bilateral cochlear removal or underwent a sham surgery. After 4, 9, or 14 days post-surgery, birds were euthanized for gene expression analysis. Bengalese finch graphic obtained from **Sainburg, 2020a**. (**C**) Example spectrograms from one hearing (sham) and one deaf (bilateral cochlear removal) bird. Songs are shown from before the procedure and 14 days following the procedure. Labels below each spectrogram correspond to discrete categories of song units ('syllables'). kHz, kiloHertz. (**D**) Uniform Manifold Approximation and Projection (UMAP) representation of syllable spectrograms (see Methods) across the entire recording period for each bird (4 days before to 14 days after the procedure). Data are split into 'pre'-procedure (4–1 day before surgery) and 'post'-procedure (1–14 days after surgery) subsets. For reference, gray points in each plot correspond to data from the other subset. Example syllable spectrograms are placed adjacent to their position in UMAP space. (**E**) Density plots of UMAP projections for the syllables from one deafened bird (shown in panel (**A**)) at two timepoints, one day before and 13 days after deafening. (**F**) Subtraction of UMAP densities in (**E**) from the average pre-procedure density. (**G**) Mean sum of UMAP density differences for syllables from birds that were either deafened (deaf) or underwent a sham surgery (hearing). For each bird and each day, positive UMAP density differences were summed and then averaged across birds. Error bands are standard errors of the mean. Color bars indicate days in which values were significantly different between deaf and hearing birds (Student's *t*-test, two-sided, p<0.05). (**H**) UMAP plot of one syllable from one deafened bird colored by the day following deafening. (**I**) Comparison of fundamental frequency (FF) variability between hearing and deafened birds. Rolling coefficient of variation (CV, window size 11 syllables) was calculated for the fundamental frequencies of each harmonic stack for each bird. Shown are two example syllables from one hearing and one deafened bird, plotted across the number of days relative to the procedure date (sham or cochlear removal). (**J**) Mean FF CV in the 7–9 days following sham or cochlear removal normalized to FF CV in the 2 days before the procedure. Linear mixed-effects regression (see Methods) was used to estimate the group post vs.

*Figure 1 continued on next page*

Figure 1 continued

pre-procedure FF CV difference for hearing and deaf birds. P-values are obtained from the regression model using Satterthwaite's degrees of freedom method.

The online version of this article includes the following figure supplement(s) for figure 1:

**Figure supplement 1.** Additional quantification of deafening-induced changes to the song.

1A–C). To visualize deafening-induced changes to a song, we calculated spectrograms for each syllable and used uniform manifold approximate projection (UMAP) to project these spectrograms onto a latent space, using an approach described in *Sainburg et al., 2020c* (*Figure 1C*, *Figure 1— figure supplement 1A–C*). Following deafening, the projections of syllable spectrograms gradually shifted to occupy different locations, indicating a change from a pre-procedure song (*Figure 1D–H*, *Figure 1—figure supplement 1A and B*). Syllable spectral changes after deafening were complex but generally trended toward an increase in syllable 'noisiness' (Wiener entropy) (*Figure 1—figure supplement 1D and E*).

Although adult birdsong is a highly precise motor skill, its features vary slightly from rendition-to-rendition, similar to other motor skills. Past work has demonstrated that this variability is in part generated by central neural mechanisms in the forebrain and is modulated by social context, indicating that song variability is an actively regulated component of birdsong (*Kao et al., 2005*; *Kao and Brainard, 2006*; *Kojima et al., 2013*; *Moorman et al., 2021*; *Sakata et al., 2008*). To assess how song variability changes following deafening, we focused on a single spectral feature, fundamental frequency (FF), and calculated its coefficient of variation (CV) across renditions (*Figure 1J and K*). Deafening resulted in a gradual increase in the CV of FF (day 7–9 post-procedure mean ± SEM, 57 ± 18%) while sham surgery elicited no change (0.11 ± 2.3%). This increase in rendition-to-rendition FF variability is consistent with reports describing an increase in within-syllable frequency modulation following deafening (*Brainard and Doupe, 2001*). These results indicate that deafening elicits both shifts in the structure of song as well as decreases in stereotypy across renditions.

## Neural circuit-wide analysis of gene expression

Birdsong is generated by a dedicated and anatomically discrete neural circuit called the song system (*Figure 2A and B*). This defined architecture allows interrogation of how the molecular and cellular properties of each region (termed 'song nuclei') influence and is influenced by birdsong performance and learning. Four primary song nuclei reside in the telencephalon: HVC (proper name), RA (robust nucleus of the arcopallium), LMAN (lateral magnocellular nucleus of the anterior nidopallium), and Area X. HVC and RA comprise the song motor pathway (SMP) and are necessary for song performance (*Nottebohm et al., 1976*; *Simpson and Vicario, 1990*). HVC influences the timing and temporal structure of song and projects to RA, which provides descending motor control of song via projections to syringeal and respiratory brainstem regions, which send recurrent connections back into the SMP to influence spectral and temporal features of the song (*Goldberg and Fee, 2012*; *Vicario, 1991*; *Wild, 1993*). HVC also projects to the striatal nucleus Area X that, together with the pallial region LMAN and thalamic region DLM, form the 'anterior forebrain pathway (AFP),' which contributes to song plasticity both during song acquisition in juveniles and song adaptation in adults (*Andalman and Fee, 2009*; *Bottjer et al., 1984*; *Brainard and Doupe, 2000*; *Charlesworth et al., 2012*; *Nordeen and Nordeen, 1993*; *Scharff and Nottebohm, 1991*; *Sohrabji et al., 1990*; *Warren et al., 2011*; *Williams and Mehta, 1999*). Each region is embedded in a larger anatomical domain that lies outside of song control circuitry but shares similar molecular, connectivity, and functional properties (*Figure 2B*; *Feenders et al., 2008*; *Helduser et al., 2013*; *Kröner and Güntürkün, 1999*). HVC is located in the dorsal part of the caudal nidopallium (NC); RA is located in the arcopallium (Arco.); Area X is located in the striatum (Stri.); and LMAN is located in the rostral nidopallium (NR). In effect, these regions serve as 'non-song' comparators for each song region that enable the identification of molecular and cellular features that are specific to song-related perturbations.

Disruptions to song that follow the loss of auditory feedback are associated with both local alterations to song nuclei as well as to the interactions among connected components of the song system neural circuit (*Brainard and Doupe, 2000*; *Hamaguchi et al., 2014*; *Kojima et al., 2013*; *Watanabe et al., 2006*). To examine how song destabilization influences gene expression in the song system

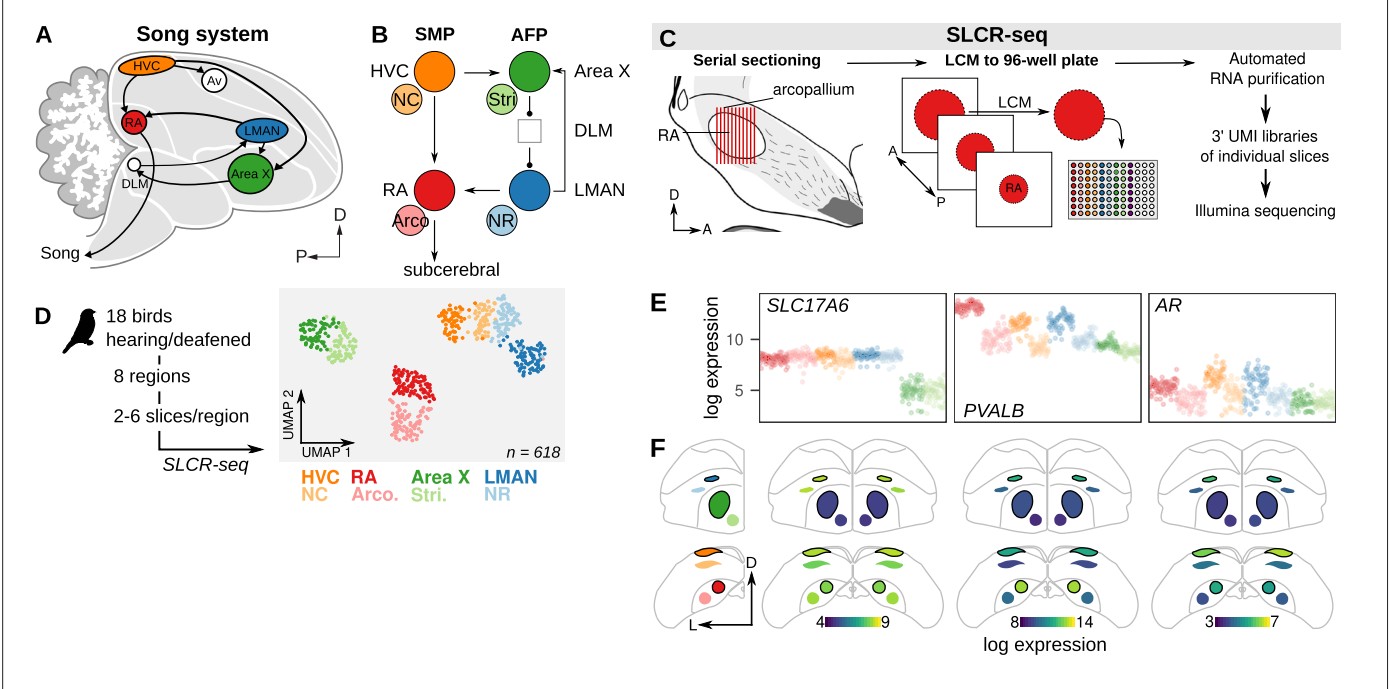

**Figure 2.** Neural circuit transcriptomics using Serial Laser Capture RNA-sequencing (SLCR-seq). (**A**) Schematic overview of the song system. HVC, proper name; RA, robust nucleus of the arcopallium; LMAN, lateral magnocellular nucleus of the nidopallium; Av, avalanche; DLM, medial portion of the dorsolateral thalamic nucleus; D, dorsal; P, posterior. (**B**) Circuit diagram of the song system. Arrowheads and closed circles indicate excitatory and inhibitory connections, respectively. NC, caudal nidopallium; Arco., arcopallium; NR, rostral nidopallium; Stri., striatum. (**C**) Schematic of SLCR-seq. Fresh-frozen brains were cryosectioned for Laser Capture Microdissection (LCM). Individual sections of regions of interest were collected into wells of 96-well plates, and then total RNA was purified using an optimized solid phase reversible immobilization (SPRI) protocol. After RNA purification, 3'-end sequencing libraries were prepared containing unique molecular identifiers (UMI) using a custom protocol. (**D**) Left: Experimental overview of SLCR-seq on hearing and deaf birds. After a baseline period of song recording, birds were either deafened through bilateral cochlear removal or underwent a sham surgery. After 4, 9, or 14 days post-surgery, birds were euthanized and SLCR-seq libraries were prepared from HVC, NC, RA, Arco., LMAN, NR, Area X, and Stri. Right: Uniform Manifold Approximation and Projection (UMAP) plot of SLCR-seq data colored by section position. Each point reflects the gene expression profile of a single SLCR-seq sample. Samples show segregation by broad anatomical area — striatal (Area X), nidopallial (HVC, NC, LMAN, NR), arcopallial (RA, Arco.) — and song system nuclei from surrounding areas. (**E**) Normalized log gene expression data of three example genes — *SLC17A6*, *PVALB*, and *AR*. Each point is gene expression in a single SLCR-seq sample. *SLC17A6* is a marker for glutamatergic cells and is distinctly depleted in the striatal samples; *PVALB* and *AR* are two genes known to be enriched in song system nuclei. (**F**) Coronal anatomical atlas representation of the expression of the three genes shown in panel (**F**). Each region is colored according to the log gene expression value. D, dorsal; L, lateral.

The online version of this article includes the following figure supplement(s) for figure 2:

**Figure supplement 1.** Additional validation of Serial Laser Capture RNA-seq (SLCR-seq) data.

at both local and circuit levels, we developed a protocol for sample collection and RNA-seq library construction that addresses three goals: (1) precise collection of histologically defined samples, (2) ease of collecting multiple replicates per animal and brain region, and (3) reduced per-sample cost for library preparation and sequencing. This approach, termed Serial Laser Capture RNA-seq (SLCR-seq), combines the anatomical precision of laser capture microdissection (LCM) with the capacity to work with large numbers of low-input RNA samples provided by single-cell RNA-sequencing protocols (*Figure 2C*, see Methods). Brains were flash-frozen without fixation and then cryosectioned onto slides suitable for LCM. We visualized song nuclei using an optimized rapid Nissl staining protocol, collected single cryosections from regions of interest in 96-well plates using LCM, then purified total RNA using a custom solid phase reversible immobilization protocol. The preparation produces high-quality RNA (RIN = 9.1 ± 0.5) and yields that are sufficient for library preparation (one 20 μm-thick section with an area of 100,000 μm² yields 1–2 ng; RA area is ~125,000 μm²). From this total RNA, we then prepared 3'-localized RNA-seq libraries containing unique molecular identifiers adapted from protocols previously developed for single-cell RNA-sequencing (*Islam et al., 2014*; *Kivioja et al., 2012*; *Picelli et al., 2014*).

We used SLCR-seq to generate RNA-seq libraries for each bird from eight brain regions — HVC, RA, LMAN, and Area X, and four paired non-song regions — (*Figure 2D*) with multiple LCM sections (2–6) collected per region per bird, yielding 598 samples after quality control filtering for the number of detected genes in each sample (mean ± s.d. number of sections per region per bird = 4.5 ± 1.5). Gene expression variation across this dataset segregated samples into three broad clusters corresponding to the region of origin — arcopallium (RA and Arco.), nidopallium (HVC, NC, LMAN, and NR), and striatum (Area X and Stri.) — consistent with the different functional properties and developmental origins of these regions (*Figure 2D*). Furthermore, these broad clusters were subdivided into adjacent but distinct song/non-song pairs, reflecting known similarities between song nuclei and adjacent neural regions (*Nevue et al., 2020*). To further validate this approach, we inspected the SLCR-seq expression values of genes with known variation across the songbird brain or enrichment in the song system (*Figure 2E and F* and *Figure 2—figure supplement 1*). The glutamatergic neuron marker *SLC17A6* was strongly depleted in the striatal regions Area X and Stri, consistent with the relative scarcity of excitatory neurons in these regions. Genes with known variation in expression across the system, e.g., parvalbumin (*PVALB*) and androgen receptor (*AR*), corroborated a strong correspondence between SLCR-seq expression values and in situ hybridization signal intensities (*Lovell et al., 2020*; *Figure 2E and F* and *Figure 2—figure supplement 1*).

## Song system-wide transcriptional signatures of song destabilization

To provide a single statistic that reflects the extent of song change for each bird, we used a previously developed method (*Mets and Brainard, 2018*) that builds statistical models of song in two conditions (e.g. pre and post-procedure) and calculates the distance between probability distributions generated from these models using Kullback-Leibler divergence ('Song $D_{KL}$,' see *Figure 3—figure supplement 1A* and Methods). Here, higher values indicate a larger divergence of post-procedure songs compared to pre-procedure songs, therefore providing a measure of song change from baseline. The UMAP quantification used in *Figure 1* was used to summarize this condensed visual representation into a single statistic. However, we choose Song $D_{KL}$ for subsequent analysis over the UMAP-based quantification because Song $D_{KL}$ is a previously validated statistical modeling approach that robustly captures alterations to a wide range of song types (*Mets and Brainard, 2018*). For each bird, we calculated Song $D_{KL}$ between songs recorded during the two days before the procedure (deafening or sham) and those recorded on the day of euthanasia and the preceding day. Deafening resulted in a significant increase in Song $D_{KL}$ relative to sham (*Figure 3A*, hearing 0.14 ± 0.041 log Song $D_{KL}$ mean ± SEM; deaf 0.52 ± 0.085 mean ± SEM, two-sided Wilcoxon rank-sum test p=5e−4). We did not include the number of days from procedure (sham or deafening) as an explicit variable in subsequent analyses since the Song $D_{KL}$ measure more directly captured the amount of alteration to song. Singing influences gene expression in the song system (*Feenders et al., 2008*; *Horita et al., 2012*; *Jarvis et al., 1998*; *Sasaki et al., 2006*; *Wada et al., 2006*; *Warren et al., 2010*; *Whitney et al., 2014*; *Whitney and Johnson, 2005*), and previous work has indicated that recent singing influences song plasticity and variability (*Chen et al., 2013*; *Hayase et al., 2018*; *Hilliard et al., 2012*; *Miller et al., 2010*; *Ohgushi et al., 2015*). Therefore, we also included terms for the number of songs sung on the day of euthanasia and the average number of songs sung per day in the pre-procedure period to control for constitutive differences in singing propensity. These values varied widely across birds (*Figure 3B*) but did not differ significantly between hearing and deafened birds (number of songs on date of euthanasia: hearing 52 ± 18 mean ± SEM, deaf 55 ± 17, two-sided Wilcoxon rank-sum test p=0.7; pre-procedure songs/day: hearing 332 ± 39 mean ± SEM, deaf 339 ± 43, two-sided Wilcoxon rank-sum test p=1).

We used multiple regression to identify genes whose expression varied with song destabilization (Song $D_{KL}$) (*Figure 3C*). A subset of hearing and deaf birds showed overlapping Song $D_{KL}$ values (three birds in each condition). To detect genes with expression differences associated with song destabilization, we compared birds in each group that had Song $D_{KL}$ values outside of this overlapping range (*Figure 3A*, six hearing birds with 'low' Song $D_{KL}$ values, and six deaf birds with 'high' Song $D_{KL}$ values). In general, birds that had been deafened for longer (9 and 14 days) had Song DKL values outside of the hearing Song $D_{KL}$ range while those deafened for less time (4 days) showed less song destabilization. Exceptions to this pattern include one 4-day-deafened bird that showed particularly strong song destabilization and one 9-day-deafened bird that showed modest song change.

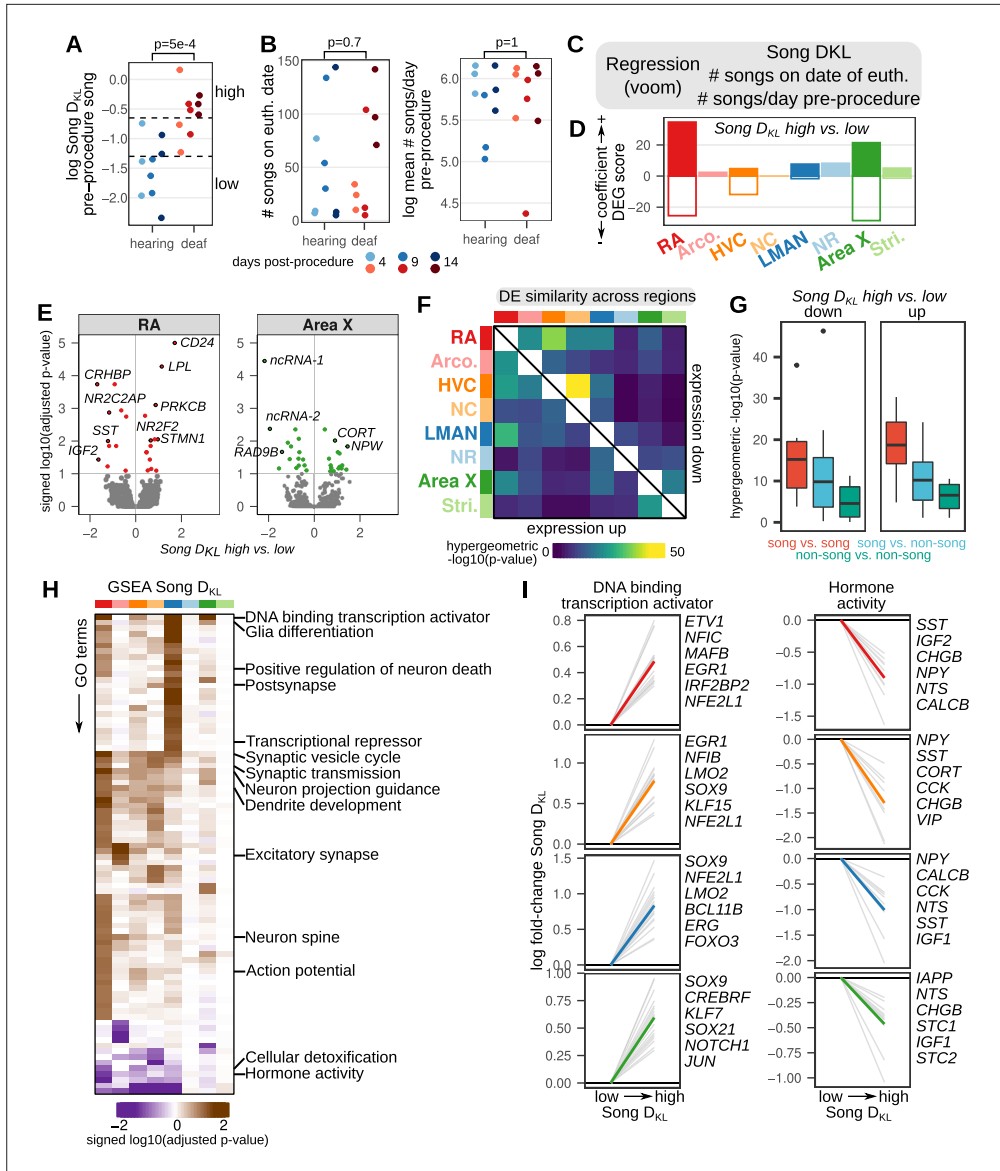

**Figure 3.** Song destabilization is associated with song system-wide alterations to gene expression. (**A**) Relative spectral distance between syllables pre- and post-procedure, represented as the mean Kullback-Leibler (KL) distance between Gaussian mixture models or 'Song $D_{KL}$' (see Methods for calculation). Song $D_{KL}$ trends higher with increasing days from deafening. Significance was calculated using a two-sided Wilcoxon rank-sum test. (**B**) (left) Number of songs sung on euthanasia date (i.e. within the two hour period between lights-on and euthanasia) and (right) log mean number of songs sung per day pre-procedure for each bird grouped by hearing or deaf. Significance was calculated using a two-sided Wilcoxon rank-sum test. (**C**) Differential expression analysis of song destabilization. Multiple regression using voom/limma provided estimates of gene expression fold change with variation in song deviation (Song $D_{KL}$), number of songs on the day of euthanasia, and baseline differences in singing rate. (**D**) Differential expression gene (DEG) scores from gene expression regressions. Positive values reflect genes with increased expression, while negative values indicate genes with reduced expression. Scores are the sum of the –1 * log10(adjusted p-values) of high vs. low Song $D_{KL}$ regression coefficients. Each score is multiplied by the sign of the coefficient to obtain a signed value. Separate coefficients were estimated for each neural region. (**E**) Volcano plots of –1 * log10(adjusted p-values) versus the log fold-change of gene expression in RA and Area X in high vs. low Song $D_{KL}$ birds. Signed adjusted p-values above five were assigned values of five to aid visualization. Labeled are the 10 genes with the highest signed adjusted p-value. 'ncRNA-1' accession is LOC116184561, 'ncRNA-2' accession is LOC116183441. (**F**) Similarity of song destabilization differential gene expression across the song system and surrounding regions. Heatmaps show the -log10(p-value) from hypergeometric tests comparing the expected versus observed overlap of the top 250 differentially expressed genes for each compared

*Figure 3 continued on next page*

*Figure 3 continued*

region, divided into genes with increased expression with song destabilization (lower left triangle) and those with decreased expression (upper right triangle). (**G**) Distribution of values from (**F**) comparing song versus song regions, song versus non-song regions, and non-song versus non-song regions. Box middle is the median, box upper and lower bounds are the 25th and 75th percentile, and whisker ends lie at 1.5 times the inter-quartile range. (**H**) Gene set enrichment analysis (GSEA) of song destabilization-associated genes. Shown are the Gene Ontology (GO) terms that are significant in at least one song or non-song region (adjusted p-value <0.1, see Methods). Heatmap represents the signed log10(adjusted p-value) for each GO term and region, with the sign indicating that a given term is associated with increased or decreased expression in Song $D_{KL}$ high versus low birds. Terms are ordered by hierarchical clustering (euclidean distance, Ward squared method). Representative terms are listed for each cluster. (**I**) Song destabilization gene expression responses of genes in two gene sets — 'DNA binding transcription activator' (GO:0001216) and 'Hormone activity' (GO:0005179) — that have differential expression across song regions. Shown are the top 20 leading edge genes from GSEA (gray lines) and the top six of these are labeled at right. The mean expression change for these 20 genes is shown as a colored line in each panel.

The online version of this article includes the following figure supplement(s) for figure 3:

**Figure supplement 1.** Extended analysis of song destabilization-associated gene expression.

To identify regional patterns of differential expression, we used a differential expression gene (DEG) score that incorporates the number of differentially expressed genes and their adjusted p-values (*Figure 3D* and *Supplementary file 2*, see Methods). Song regions generally showed greater expression differences than non-song regions for both Song $D_{KL}$ and singing-rate (*Figure 3D* and *Figure 3—figure supplement 1B, C*). Across the song regions, the largest changes to gene expression between high and low Song $D_{KL}$ occurred for the forebrain motor output nucleus RA and the striatal component of the song system Area X (*Figure 3D*, red and green bars). For these regions, the most significantly modulated genes (adjusted p-value <0.1) were equally likely to be upregulated versus downregulated in deafened versus hearing birds (*Figure 3E*; for RA, 14 genes upregulated and 11 genes downregulated; for Area X, 15 genes up, 16 genes down). Among the most highly upregulated genes in RA were the plasticity-associated gene protein kinase C β (*PRKCB*) (*Chu et al., 2014*; *Fioravante et al., 2014*), whose protein levels were previously shown to be upregulated in RA following deafening (*Watanabe et al., 2002*); the microtubule-destabilizing protein stathmin 1 (*STMN1*), which has roles in long term potentiation and fear memory formation (*Shumyatsky et al., 2005*); *CD24*, a surface protein that influences neurite extension (*Gilliam et al., 2017*); and the lipid processing enzyme lipoprotein lipase (*LPL*), which is implicated in memory formation and Alzheimer's disease pathology (*Wang and Eckel, 2012*; *Yu et al., 2015*). Likewise, among the most downregulated genes in RA were secreted neuromodulatory proteins including corticotropin-releasing hormone binding protein (*CRHBP*), somatostatin (*SST*), and insulin growth factor 2 (*IGF2*), which each have described roles in regulating neuronal physiology and neural plasticity (*Chen et al., 2011*; *Li et al., 2016*; *Song et al., 2021*).

To further examine the neural-circuit-wide structure of gene expression across the song system and surrounding regions, we pairwise intersected the lists of the top Song $D_{KL}$ differentially expressed genes (250 genes with the lowest p-values) for each region and calculated the degree of overlap using a hypergeometric test (*Figure 3F and G*). Song destabilization-associated differential gene expression was more similar between song regions than between both song and non-song pairs and non-song regions with each other (*Figure 3G*), indicating that the song system exhibits, in part, a common transcriptional response to song destabilization that is not shared in adjacent regions. We performed gene set enrichment analysis (GSEA) of differentially expressed genes to identify pathways that show coherent gene expression responses to song destabilization (*Figure 3H* and *Supplementary file 3*). Several pathways exhibited similar expression responses across all four song regions, including those related to transcription regulation, glia differentiation, and hormone activity (*Figure 3H1*). Genes related to synaptic transmission were differentially expressed across multiple pallial regions, including song regions RA, HVC, and LMAN as well as the non-song region NCL. Neuron spine-associated genes were upregulated across RA, Arco., and HVC, consistent with previous reports of altered spine dynamics in the song motor pathway following deafening (*Peng et al., 2012a*; *Peng et al., 2013*; *Tschida and Mooney, 2012*; *Zhou et al., 2017*).

## Correlated gene modules associated with song destabilization

The regression analysis described in *Figure 3* identified differential expression at the level of individual genes but may have missed subtler expression responses that are correlated across multiple genes. To better identify groups of differentially expressed genes with similar responses to song destabilization, we leveraged the large sample numbers of the SLCR-seq dataset to perform gene-gene correlation network analysis for each region separately using MEGENA (Multiscale Embedded Gene Co-expression Network Analysis), an approach that generates sparse networks of covarying genes by applying a topological constraint to co-expression networks (*Song and Zhang, 2015*; *Figure 4A* and *Figure 4—figure supplement 1*). Using this method, we constructed gene-gene correlation networks for each song and non-song region separately, combining SCLR-seq samples across all birds, both hearing and deafened. Mapping song destabilization fold-change of each gene onto the RA network showed a segregation between genes with increased and decreased expression, indicating that expression differences associated with song destabilization state are prominent drivers of network structure (*Figure 4B*). This segregation was also seen for the HVC, LMAN, and Area X networks (*Figure 4—figure supplement 1A*). MEGENA employs a hierarchical module detection algorithm to identify correlated sets of genes at different levels of resolution (*Supplementary file 4*). For each module in each region's network, we averaged high-vs-low Song $D_{KL}$ regression coefficients for its member genes and compared these observed mean values to a shuffled distribution of mean values, generated by sampling the same number of module genes across the network at random. Several modules in each region's network showed significantly higher or lower mean fold-change relative to a distribution of mean fold-changes from sets of randomly selected genes (100 random samplings of genes from the network, shuffled p-value <0.01, *Figure 4C* and *Figure 4—figure supplement 1B*).

To assess the similarity between modules in the correlation networks of one brain region and those in the networks of other brain regions, we calculated a module preservation score (see Methods) and found that RA destabilization-associated modules were preserved to different degrees in networks for the other song regions. In addition, several RA modules showed similar response patterns in other song regions, such that modules upregulated in RA were upregulated in HVC, LMAN, and Area X and likewise for downregulated modules (*Figure 4D and E* and *Figure 4—figure supplement 1B*). This pattern is consistent with the overall similarity in differential expression seen among song regions using the regression analysis described in *Figure 3* (*Figure 3F*). Gene set enrichment analysis indicated that differential RA modules are enriched for a range of biological pathways (*Figure 4F* and *Figure 4—figure supplement 1C*). Notably, the top downregulated module (M74) was enriched for secreted proteins, such as *CRHBP*, *SST*, and *CHGB* (*Figure 4G*). Upregulated modules were enriched for genes involved in development, morphogenesis, and gene regulation, including *PBX1*, *NR2F2*, *ZHX3*, and *ANKRD11*.

## Cell type expression of song destabilization-associated genes

After establishing a circuit-wide view of gene expression responses to song destabilization, we investigated the cellular specificity of these responses to understand what cell classes exhibit the most substantial transcriptional changes and may play a role in deafening-induced song plasticity. To do so, we integrated the SLCR-seq data with a previously generated single-nucleus and single-cell RNA-sequencing dataset from HVC and RA of hearing adult male finches (*Colquitt et al., 2021*; *Figure 5A*). In that work, we compared songbird neuronal classes in HVC and RA to those in mammals and identified a high degree of transcriptional similarity across several neuronal classes (*Figure 5B*).

For each gene, we computed a cell type destabilization score — the product of a gene's cell type specificity with its fold-change between high and low Song $D_{KL}$ — to assay cellular biases of destabilization-associated expression (*Figure 5C and D*, and *Figure 5—figure supplement 1A, B*). In RA, which showed the strongest transcriptional changes as described above (*Figure 3D*), differentially expressed genes were most strongly localized to neurons. In particular, genes with reduced expression during song destabilization, such as *CRHBP*, *SST*, *NPY*, and *CHGB*, showed a bias toward *Sst*- and *Pvalb*-class interneurons (GABA-2/3/4). In addition, several upregulated genes, such as *PRKCB*, *EPHB1*, and *DNM1*, were biased toward RA glutamatergic neurons. HVC showed a similar pattern of cell-type expression, with genes that had reduced expression biased toward *Sst*-class interneurons as well as LGE-class GABA-1 and MGE-class GABA-7 interneurons (*Figure 5—figure supplement 1A and B*). These cellular expression biases could arise from increases or decreases in the abundances of

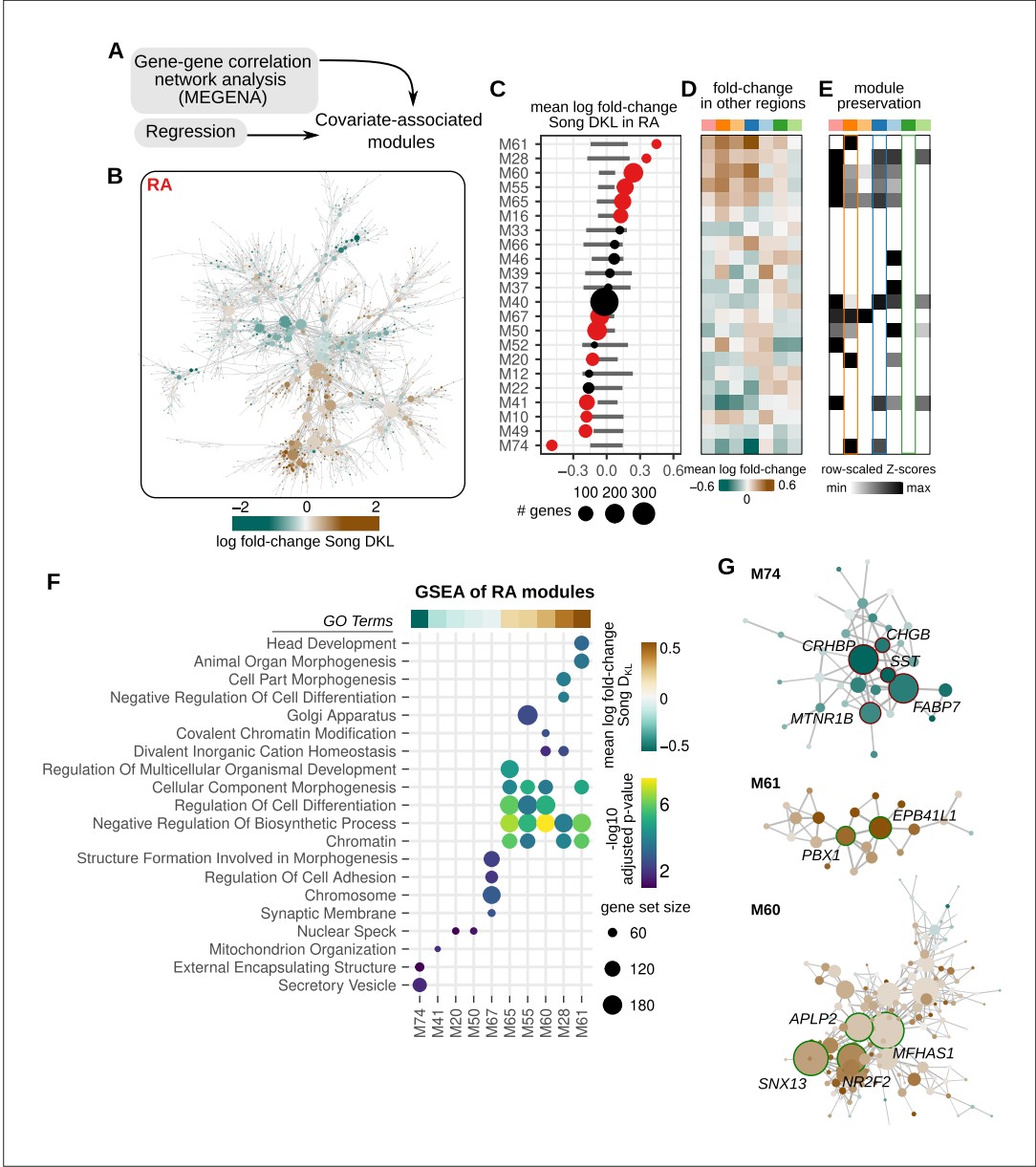

**Figure 4.** Correlated modules of gene expression associated with song destabilization. (**A**) To identify correlated patterns of gene expression, gene-gene correlation networks were constructed for each region using MEGENA (Multiscale Embedded Gene Co-expression Network Analysis). These networks were then used to identify correlated sets of gene modules. Estimated regression coefficients were mapped onto correlation networks to identify covariate-associated expression modules. (**B**) Gene-gene correlation network for RA. Each node is colored by the log fold-change expression between deaf and hearing birds. (**C**) Average song destabilization gene expression changes for each RA module. Error bars are null distributions generated by repeatedly sampling the network (100 times) for the number of nodes in a given module and then averaging their high vs. low Song $D_{KL}$ fold-changes. Dots that are colored have mean coefficient values that are lower or higher than 1% or 99% of the sampled distribution, respectively. (**D**) Average change in expression of RA modules across each song system and non-song system region. (**E**) Preservation scores for RA modules in the correlation networks of other song and non-song system regions. Only significant values are shown (Bonferroni p-values <0.01), and values are scaled to the maximum and minimum for each module to show relative levels of preservation across regions. (**F**) Gene set enrichment analysis (GSEA) of RA modules with significant gene expression alteration with song destabilization. Mean fold-change values for each module, as represented in (**B**), are shown at the top of the GSEA plot. Shown are at most the top five significant Gene Ontology (GO) terms (GSEA adjusted p-values <0.2). (**G**) Network diagrams for three modules (M61, M60, and M74) that show large deviations with song destabilization. Labeled

*Figure 4 continued on next page*

*Figure 4 continued*

and highlighted are selected hub genes for each module (see Methods for classification). Node colors indicate log fold-change expression between deaf and hearing birds (scale given in (**A**)).

The online version of this article includes the following figure supplement(s) for figure 4:

**Figure supplement 1.** Network analysis of song destabilization-associated gene expression.

defined cell populations. To determine if these cellular biases do reflect a bulk loss of particular cell types, we analyzed the differential expression of marker genes for each cell type (*Figure 5—figure supplement 1C*). In both song nuclei, markers for each neuronal cell type (see Methods for definition) showed no significant difference between high and low Song $D_{KL}$ conditions (median of fold-changes

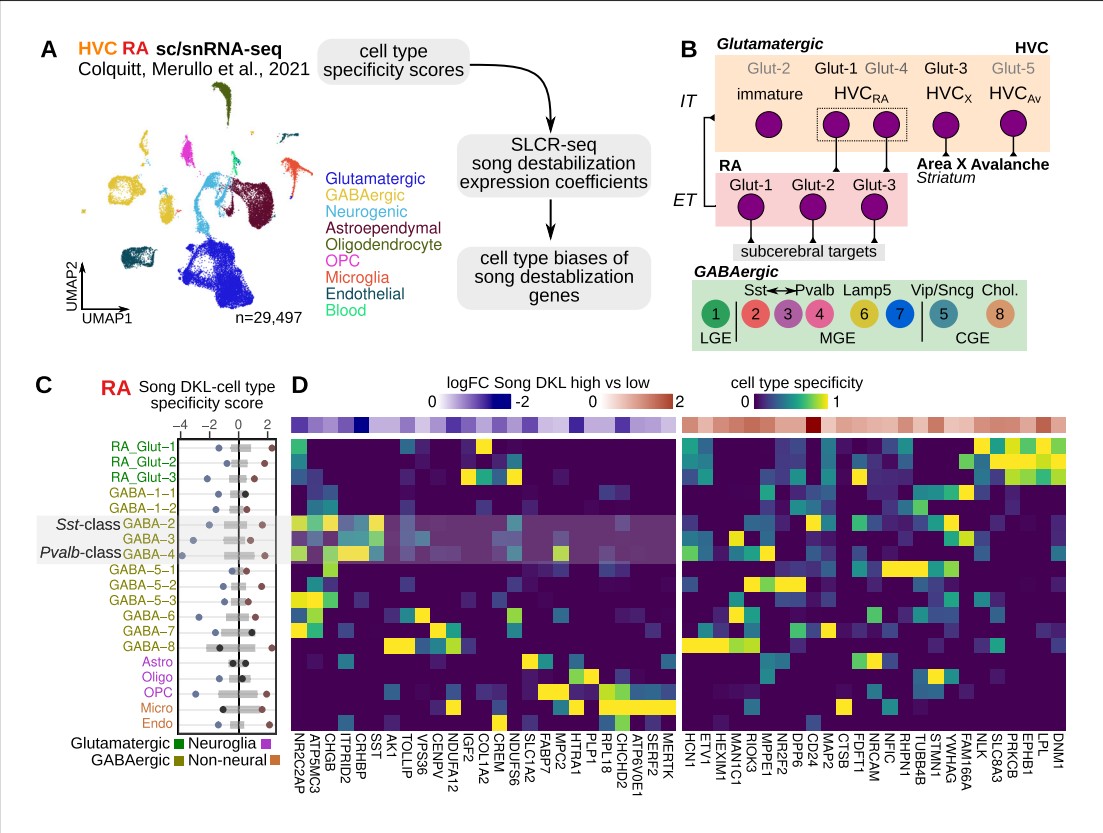

**Figure 5.** Cell-type specificity of destabilization-modulated genes. (**A**) Schematic of the approach to determine the cell type expression biases of genes that are differentially regulated with song destabilization. A previously generated cell-resolved gene expression dataset for RA (robust nucleus of the arcopallium) and HVC (proper name) (*Colquitt et al., 2021*) was combined with RA and HVC song destabilization regression coefficients from this study to compute a cell-type bias score (see Methods). Shown also is a uniform manifold approximate projection (UMAP) plot of the full dataset with major cell type groups indicated. OPC, oligodendrocyte precursor cell. (**B**) Schematic of neuronal cell types in the song motor pathway, as previously defined in *Colquitt et al., 2021*. HVC glutamatergic neurons are broadly similar to intratelencephalic (IT) mammalian neocortical neurons from multiple layers, and RA neurons are similar to extratelencephalic (ET) neurons from layer 5. Eight primary GABAergic clusters are found equally in both HVC and RA and are organized into clusters corresponding to subpallial regions of origin — lateral, medial, and caudal ganglionic eminences (LGE/MGE/CGE). The LGE-class GABA-1 has no known correspondence with mammalian neocortical neurons; GABA-2 is transcriptionally similar to Sst-class neurons, GABA-4 is similar to Pvalb-class neurons, and GABA-3 is transcriptionally intermediate between GABA-2/4. (**C**) Integration of cell type specificity scores and song destabilization differential expression to identify cell type-associated transcriptional effects of song destabilization. For the top 50 differentially expressed genes, cell type specificity was multiplied by log fold-change between high and low Song $D_{KL}$ birds. These values were then split by sign then summed within each cell type to yield a cell type Song $D_{KL}$ score. Gray bars indicate the distribution (1–99%) of Song $D_{KL}$-cell type specificity scores for 100 random sets of 50 genes. (**D**) RA cell type specificity scores for top Song $D_{KL}$ differentially expressed genes, divided into upregulated and downregulated genes. Values are scaled for each gene such that the cell type with the highest specificity score equals 1 and that with the lowest equals 0. At the top of each specificity score heatmap is the log fold-change expression for high vs. low Song $D_{KL}$.

The online version of this article includes the following figure supplement(s) for figure 5:

**Figure supplement 1.** Cell type-associated differential expression.

greater than 99% or less than 1% of medians from randomly selected genes), indicating that the cell type biases of destabilization associated-genes are not due to changes in cell type abundance, but rather to the expression levels of specific genes within defined cell classes.

## Inter-region correlation of gene expression is reduced in deafened birds

The foregoing analyses focused on comparing gene expression responses to deafening that are local to each region. However, the song system is an interconnected neural circuit, and gene expression in one region could be correlated with that in others due to shared patterns of neural activity, common responses to hormonal signaling, or to baseline expression differences across regions that vary in a concerted fashion across individuals. By similar logic, manipulations such as deafening that could disrupt global patterns of neural or hormonal signaling might result in alterations in the patterns of inter-region correlations in gene-expression.

To determine whether and how deafening alters inter-region correlations in gene expression, we first identified genes that have correlated expression between brain regions across birds. Briefly, for each gene, we calculated correlation values for each pairwise combination of brain regions, yielding region-by-region correlation matrices (*Figure 6A*). We identified significant correlations as those that were less than the 2.5% quantile or greater than the 97.5% quantile of a shuffled distribution (see Methods). We calculated the across-bird gene expression similarity between regions as the number of thresholded correlations. This analysis revealed several notable relationships among brain regions. First, each song nucleus had the highest gene correlation strength with its paired non-song region (with the exception of HVC which had generally weaker correlation strengths with other regions), consistent with the shared molecular profiles of each song nucleus with the surrounding tissue. Second, the nuclei of the vocal motor pathway (HVC and RA) and anterior forebrain pathway (LMAN and Area X) were more correlated with each other than with nuclei in the other pathway. Third, normalizing correlation strength for each song nucleus recovered known connections between nuclei (*Figure 6B*): HVC displayed strong correlations with its target RA, and LMAN was strongly correlated with both of its direct targets, RA and Area X. Interestingly, we found relatively weaker gene correlation strength between HVC and its target Area X.

To identify genes that have shared patterns of inter-region correlation across multiple song nuclei, we next clustered genes by the similarity of the correlations between song nuclei known to be directly connected, HVC-RA, LMAN-RA, HVC-X, and LMAN-X (*Figure 6C*). This analysis generated a diversity of patterns with most genes showing correlated expression among the three pallial song nuclei, HVC-RA and LMAN-RA (cluster 3). Gene set enrichment analysis indicated that this cluster is enriched for genes that are associated with signaling receptor binding and that are responsive to neural activity (*Figure 6D*). Indeed, the genes most strongly associated with HVC-RA and LMAN-RA correlations included the activity-dependent genes *CRHBP*, *NR4A3,* and *NRN1* (*Figure 6E*).

We then assessed how deafening alters gene expression correlations across the song system. To do so, we computed pairwise correlations for each gene between each region for hearing and deaf birds separately, then computed a differential matrix comparing absolute correlations in deaf birds to those in hearing birds (*Figure 6F–H*). Differentially correlated genes were defined as those with a deaf versus hearing value less than (decorrelation) or greater than (correlation gain) the extreme values of a shuffled distribution calculated for each pairwise comparison (2.5% or 97.5%, respectively). Overall, each directly connected pair of song regions had a greater number of genes with reduced correlation in deaf versus hearing birds than increased correlation (*Figure 6F and G* and *Supplementary file 5*). Two of the most strongly decorrelated genes highlight this effect. Expression of the neurotrophic factor *BDNF* was positively correlated between LMAN and RA in hearing birds but was uncorrelated in deaf birds; similarly, expression of the nuclear receptor *PPARG* was negatively correlated between LMAN and Area X in hearing birds but was uncorrelated in deaf birds (*Figure 6H*).

## Loss of afferent input to the motor pathway affects the expression of song destabilization-associated genes

The output nucleus of the anterior forebrain pathway, LMAN, is required for adaptive plasticity to song and moment-by-moment song variability and is one of the two major afferents to the motor nucleus RA (*Andalman and Fee, 2009*; *Kao et al., 2005*; *Nottebohm et al., 1982*; *Olveczky et al., 2005*;

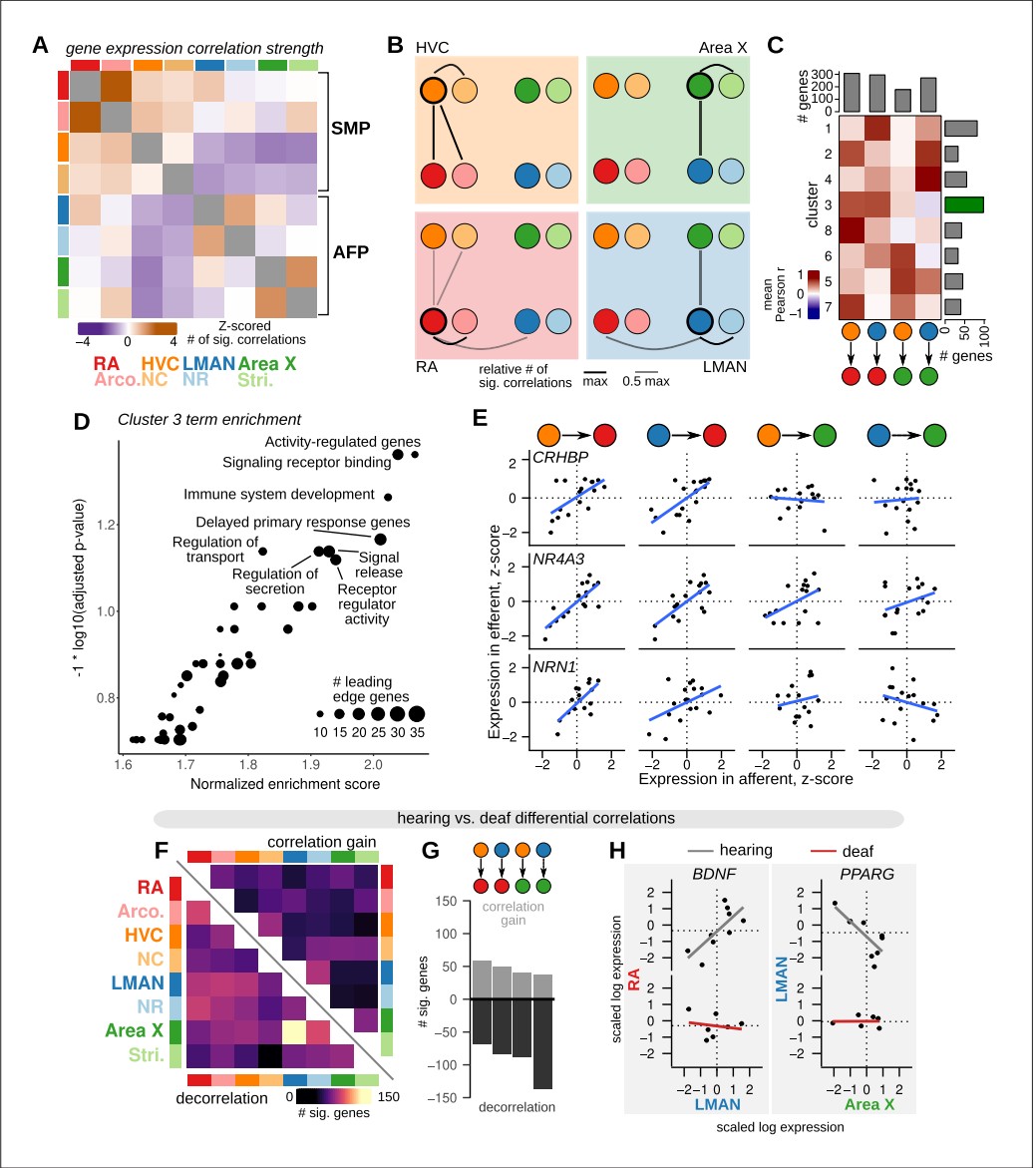

**Figure 6.** Inter-region gene expression correlation and decorrelation with song destabilization. (**A**) Inter-region gene expression correlations across song and non-song regions. For each region, the 500 genes with the highest variability across birds were selected (see Methods), then the expression of each gene was correlated across regions. Significant genes were called as those with an observed correlation less than 1% or greater than 99% of a shuffled correlation distribution (100 shuffles, calculated for each pairwise comparison between regions). The number of significantly correlated genes was Z-scored across the set of pairwise comparisons to highlight the relative strength of inter-region expression correlations. Within-region correlations were excluded from the Z-scoring and are colored gray. (**B**) Representation of the data in (**A**) showing the strength of gene expression correlation between each song system nucleus and other assayed regions. (**C**) Patterns of inter-region gene expression correlations. Genes were clustered into eight clusters by their pairwise correlation values between HVC-RA, LMAN-RA, HVC-Area X, and LMAN-Area X. Heatmap shows mean correlations within each cluster (rows) and region comparison (columns). Barplots represent the number of genes in each row or column. Highlighted is cluster 3, the HVC-RA and LMAN-RA correlation cluster, which has the greatest number of genes. (**D**) Gene set enrichment analysis (GSEA) indicates that cluster 3 is enriched for genes that are activity-dependent and have a signaling-related function. (**E**) Expression of three cluster 3 genes across pairs of song system regions with direct projections — HVC to RA, LMAN to RA, HVC to Area X, and LMAN to Area X. Each point is the z-scored expression estimate for each nucleus in one bird. (**F**) Schematic of inter-region gene expression differential correlation analysis between hearing and deaf birds. For each gene, region pair, and condition, a Pearson correlation was calculated, then a differential correlation was calculated as the difference between unsigned correlations for hearing versus

*Figure 6 continued on next page*

*Figure 6 continued*

deaf conditions. To determine significance, 100 random permutations of the expression data were made for each gene, region pair, and condition and differential correlation was computed in the same manner as for the observed values. Genes with observed differential correlations in the top or bottom 2.5% of the shuffled distribution were considered significant. (**G**) Analysis of inter-region gene expression correlations compared between hearing and deaf birds. Heatmap shows the number of genes that became decorrelated (bottom-left) or gained correlation (top-right) in deaf versus hearing birds. (**H**) Total number of genes that show significant correlation gain or decorrelation in the four pairs of assayed song system regions with direct projections. (**I**) Examples of genes that show decorrelation in deafened birds relative to hearing birds. BDNF (brain-derived neurotrophic factor) expression is correlated between RA (y-axis) and LMAN (x-axis) in hearing birds but shows no inter-region correlation in deafened birds. Similarly, PPARG expression is correlated between LMAN (y-axis) and Area X (x-axis) in hearing birds but is uncorrelated in deaf birds. Each point is the z-scored expression estimate for one bird.

*Warren et al., 2011*; *Williams and Mehta, 1999*; *Figure 2A and B*). Lesions of this nucleus result in reduced song variability (*Kao and Brainard, 2006*) and, when performed before cochlear removal, prevent song destabilization (*Brainard and Doupe, 2000*), indicating that deafening generates plasticity signals that require inputs from LMAN. We hypothesized that lesions of LMAN would establish a molecular state in RA similar to that found in other low variability and low plasticity conditions, such as that in normal hearing adult birds (versus deaf birds). To assess LMAN's influence on gene expression in the song motor pathway, we unilaterally lesioned LMAN in five adult male birds (*Figure 7A* and *Figure 7—figure supplement 1*). Unilateral LMAN lesions did not grossly alter song, and song stability as measured by Song $D_{KL}$ was similar to that for unlesioned hearing birds (*Figure 7—figure supplement 1*). Sixteen days after lesioning, we collected HVC, NCL, RA, Arco., and the primary auditory area Field L for SLCR-seq (*Figure 7B*, 91 libraries total). Field L, a region that is easily identifiable using the rapid Nissl stain protocol used in SLCR-seq, was added here to provide a control region that was outside of the song motor pathway. Song regions from each hemisphere were collected independently to allow within-bird comparisons between regions ipsi- and contralateral to the lesion. LMAN was not substantially lesioned in one bird (lesion extent ~0% of LMAN volume, see Methods), and samples from each hemisphere for this bird were treated as unlesioned. Unlike mammals, birds do not have an interhemispheric connection at the level of the forebrain, such that there is no direct connectivity between song system nuclei across hemispheres (*Nottebohm et al., 1982*; *Nottebohm et al., 1976*). We reasoned that gene expression modulated directly by LMAN activity would show specific effects in its direct target RA relative to regions that do not receive direct afferents from LMAN such as HVC and surrounding regions that are not part of the song system (Arco., NCL, and Field L).

For each brain region, we performed comparisons between the region ipsilateral to the LMAN lesion to that in the contralateral hemisphere (*Figure 7C* and *Supplementary file 6*). As expected, RA exhibited the greatest expression changes between sides ipsilateral and contralateral to the lesion (35 genes with reduced and 40 genes with increased expression in ipsilateral, adjusted p-value <0.1) compared to ipsilateral to contralateral comparisons of non-direct targets of LMAN (*Figure 7C and D*). Genes that were more highly expressed ipsilateral to the lesion were enriched for immune-responsive genes likely reflecting an injury response in RA to the afferent lesion (*Figure 7D and E*). In contrast, genes with reduced expression ipsilateral to the lesion were enriched for a range of biological processes, including activity-dependent delayed primary response genes (*Tyssowski et al., 2018*), neuron cellular homeostasis, metalloendopeptidase activity, and potassium channels (*Figure 7E*).

To examine more broadly how these expression alterations compared to those associated with deafening-induced song destabilization, we calculated the average ipsilateral versus contralateral fold change for the destabilization-associated gene modules described in *Figure 4* (*Figure 7F*). If LMAN lesions impose a molecular state associated with low variability and low plasticity, we would expect to see an inverse pattern of expression between Song $D_{KL}$ and lesion differential expression. Indeed, on the whole, modules that had increased expression in RA with higher Song $D_{KL}$ had lower expression ipsilateral to the lesion, and vice versa (*Figure 7G*). However, one module, M74, showed an opposite pattern — it was the most strongly reduced module both with increased song destabilization and with LMAN lesions. M74 hub genes *CRHBP* and *SST* were reduced specifically in RA ipsilateral to the lesion and showed no change in other assayed regions (*Figure 7H* and *Figure 7—figure supplement 2A–C*). This module is enriched for secreted neuropeptides, and the similarity of its expression change

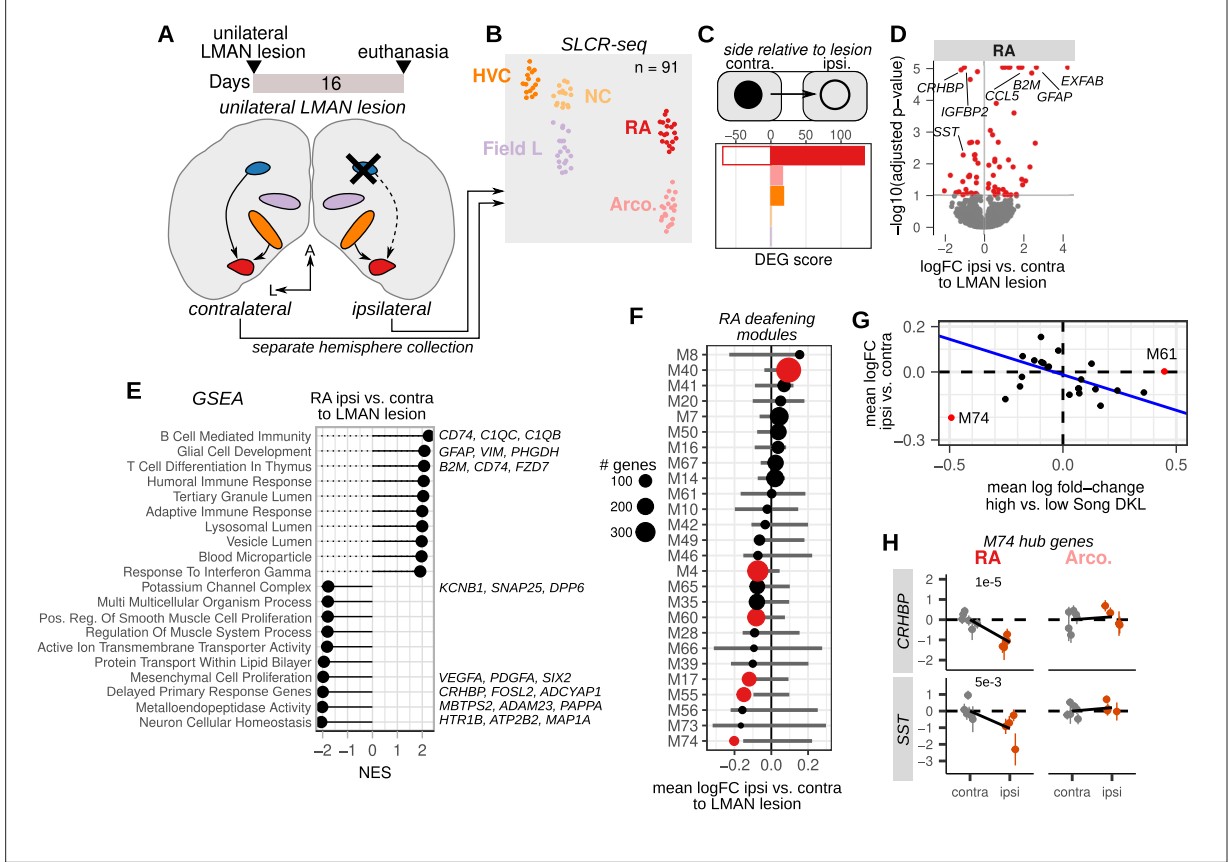

**Figure 7.** Loss of afferent input to the motor pathway nucleus RA (robust nucleus of the arcopallium) alters destabilization-associated gene expression. (**A**) Schematic of unilateral LMAN (lateral magnocellular nucleus of the anterior nidopallium) lesions and sample collection for Serial Laser Capture RNA-seq (SLCR-seq). Five birds received unilateral LMAN lesions (three left and two right hemisphere). After 16 days, birds were euthanized, and HVC (proper name), NC, RA, arcopallium (Arco.), and the primary auditory region Field L were collected for SLCR-seq. Each hemisphere was processed separately to examine the ipsilateral versus the contralateral influence of LMAN lesioning on gene expression in hearing birds. (**B**) UMAP plot of SLCR-seq samples colored by region. (**C**) Within-bird differential expression analysis of the influence of LMAN lesions on different regions of the songbird brain. Shown are differential expression gene (DEG) scores for each region assayed. Positive values reflect genes with increased expression, while negative values indicate genes with reduced expression. DEG scores are calculated as the sum of the –1 * log10(adjusted p-values) of regression coefficients for gene expression in brain regions ipsilateral versus contralateral to the LMAN lesion. Each score is multiplied by the sign of the coefficient to obtain a signed value. Separate coefficients were estimated for each neural region. (**D**) Volcano plot showing the genes that had the most significant difference in expression (red points) between RA on the ipsilateral versus the contralateral side of LMAN lesion, quantified as –1 * log10(adjusted p-values) versus the log fold-change of gene expression for RA ipsilateral versus contralateral to the LMAN lesion. Signed adjusted p-values above five were assigned values of five to aid visualization. (**E**) Gene set enrichment analysis of differential expression in RA ipsilateral versus contralateral to the LMAN lesion side. Top leading edge genes are listed at right. NES, normalized enrichment score. (**F**) Average expression change between ipsilateral and contralateral RA for each Song $D_{KL}$ module that was identified in *Figure 4*. Error bars are null distributions generated by repeatedly sampling the network (100 times) for the number of nodes in a given module and then averaging their differential ipsilateral versus contralateral coefficients. Dots are colored that have mean coefficient values that are lower or higher than 1% or 99% of the sampled distribution, respectively. (**G**) Comparison of mean fold-change expression differences in RA between high-vs-low Song DKL and contralateral versus ipsilateral to LMAN lesions. Blue line indicates linear regression through the data after excluding outlier modules M74 and M61. (**H**) Expression of two M74 hub genes, *CRHBP* and *SST*, between RA and Arco contralateral or ipsilateral to the LMAN lesion. Each dot is the estimated gene expression within a given bird and region, and error bars are standard errors of this estimate. Adjusted p-values were obtained from the ipsilateral versus contralateral regression analysis.

The online version of this article includes the following figure supplement(s) for figure 7:

**Figure supplement 1.** Validation and quantification of unilateral lateral magnocellular nucleus of the anterior nidopallium (LMAN) lesions.

**Figure supplement 2.** Validation of unilateral lateral magnocellular nucleus of the anterior nidopallium (LMAN) lesion effects on robust nucleus of the arcopallium (RA) gene expression.

across both deafening and LMAN lesions could reflect its sensitivity to altered neural activity in RA, either through the loss of auditory input or through the direct loss of a major afferent to RA.

We integrated ipsi-vs-contralateral lesion differential expression with cell type specificity, as described above in the deafening analysis, to examine the cellular expression biases of genes that are influenced by the loss of LMAN. Upregulated genes were primarily expressed in non-neuronal cells and in particular in microglia and astrocytes, consistent with an injury response (*Figure 7—figure supplement 2D and E*). Indeed, marker genes for microglia show a strong increase in expression in RA ipsilateral to LMAN lesions, suggesting an increase in microglia abundance in RA (*Figure 7—figure supplement 2F*). This bias is consistent with a glial injury response to the lesion or alternatively may reflect glia-mediated synaptic plasticity. In contrast, downregulated genes were largely expressed in neurons (*Figure 7—figure supplement 2D and E*), namely glutamatergic projection neurons (Glut-1) and MGE-derived GABAergic interneurons such as *Sst*-class (GABA-2), *Pvalb*-class (GABA-4), and cholinergic neurons (GABA-8).

## Discussion

Sensory feedback is necessary for the reliable and successful execution of learned motor skills, and its loss can lead to increased motor errors and aberrant motor plasticity. The deprivation of sensory experience has been used effectively to characterize plasticity within sensory systems and its underlying cellular and molecular mechanisms. In contrast, how sensory deprivation drives plasticity in associated sensorimotor circuitry at cellular and molecular levels is comparatively poorly understood. Here, we used the experimental advantages of birdsong, a highly precise learned motor skill that has a dedicated neural circuitry, to identify molecular pathways in sensorimotor circuits that are influenced by the loss of auditory input and associated vocal motor destabilization.

This model has particular relevance for understanding the neural basis of speech alterations caused by deafening that occurs after speech acquisition (post-lingual deafening). Similar to the effects of auditory feedback loss to birdsong, post-lingual deafening in humans reduces the rendition-to-rendition precision of speech production (*Lane and Webster, 1991*; *Waldstein, 1990*) and alters spectrotemporal features of speech (*Lane and Webster, 1991*; *Schenk et al., 2003*). Finally, the deterioration of both speech and birdsong is more extreme when deafening occurs at earlier ages, suggesting that there are similar age-dependent mechanisms of vocal stabilization in both systems (*Brainard and Doupe, 2001*; *Cowie et al., 1982*; *Lombardino and Nottebohm, 2000*; *Waldstein, 1990*). It is an open question how the different components of speech production neural circuitry respond to hearing loss at various biological levels, from molecular to physiological.

For the songbird, prior studies have identified a variety of circuit, cellular, and molecular mechanisms that may contribute to deafening-induced song-destabilization (*Brainard and Doupe, 2000*; *Kojima et al., 2013*; *Mandelblat-Cerf et al., 2014*; *Mori and Wada, 2015*; *Peng et al., 2012a*; *Peng et al., 2013*; *Peng et al., 2012b*; *Scott et al., 2000*; *Tschida and Mooney, 2012*; *Wang et al., 1999*; *Watanabe et al., 2002*; *Zhou et al., 2017*). These prior demonstrations, which have focused on a disparate set of song control structures and specific candidate mechanisms, motivated our interest in applying a circuit-wide and unbiased approach in this system to identify molecular responses to auditory deprivation-induced motor destabilization. Understanding these responses in the songbird vocal control system could provide insight into the neural mechanisms underlying the plasticity and resilience of both learned vocalizations and other well-learned motor skills.

### Molecular localization of song destabilization

Past work on the neural mechanisms underlying song plasticity has largely focused on changes occurring in one or two brain regions at a time. Song destabilization in adult songbirds is associated with a variety of changes to the morphology and physiology of neurons in the song system including changes to dendritic spine stability and synapse densities in HVC and RA (*Tschida and Mooney, 2012*; *Zhou et al., 2017*); alterations to song tuning responses in LMAN (*Roy and Mooney, 2007*) and decreased synaptic inputs onto and increased intrinsic excitability of HVC projection neurons (*Hamaguchi et al., 2014*; *Tschida and Mooney, 2012*). Our neural circuit-wide analysis of gene expression responses to deafening allowed us to investigate which regions of the song system show the strongest transcriptional changes during song destabilization, providing a readout of the molecular correlates of neural

plasticity. We found that the motor output nucleus RA showed the highest differential expression upon song destabilization, with substantial changes also found in Area X and to a lesser extent in HVC and LMAN. RA lies at the nexus of the motor pathway and the anterior forebrain pathway, a circuit required for song plasticity, and is a major locus of neural plasticity during juvenile song learning and adult song adaptation (*Garst-Orozco et al., 2014*; *Miller et al., 2017*; *Ölveczky et al., 2011*). This position in the song neural circuit makes it well situated to integrate neural activity associated with the stable motor program with AFP-generated contributions to deafening-induced plasticity.

Some of the most salient pathways upregulated in RA were associated with synaptic transmission and neuron spines, consistent with previous reports that found that deafening increases synapse densities, spine densities, and spine lengths in RA (*Peng et al., 2012a*; *Zhou et al., 2017*). These terms were also enriched to some extent for differentially expressed genes in HVC, in which neurons projecting to Area X exhibit decreased spine stability following deafening (*Tschida and Mooney, 2012*). Past work on several of the top differentially expressed genes in RA supports a general role for altered synapse and spine dynamics during deafening-induced song plasticity. In particular, stathmin 1 (*STMN1*) is located at synapses and binds to tubulin to inhibit microtubule formation (*Curmi et al., 1997*; *Shumyatsky et al., 2005*). Knockout of *STMN1* in mice results in impaired long-term potentiation in the amygdala and reduced memory in fear-conditioning tasks (*Shumyatsky et al., 2005*). Furthermore, *STMN1* is differentially phosphorylated during fear conditioning, altering its activity and AMPA receptor localization to the synapse (*Uchida et al., 2014*). Similarly, the surface glycoprotein *CD24*, which is upregulated in RA, influences neurite outgrowth (*Gilliam et al., 2017*) as well as synapse formation and transmission (*Jevsek et al., 2006*). Lastly, the lipid processing enzyme lipoprotein lipase (*LPL*) is upregulated in RA during song destabilization. Knockouts of *LPL* in mice result in impaired learning and memory, decreased presynaptic vesicles in the hippocampus (*Xian et al., 2009*), and reduced AMPA receptor expression (*Yu et al., 2015*).

The expression of neuropeptides was also broadly reduced following deafening across multiple song nuclei. This result suggests that song plasticity is a product of not only alterations to synapse structure and neurotransmitter-mediated signaling but also changes in neuromodulation. It has not been well-examined how secreted neuropeptides influence birdsong plasticity and neural activity in birdsong neural circuitry. However, extensive evidence garnered in other systems indicates that neuropeptide signaling has a powerful effect on neural circuit activity, plasticity, and behavioral output (*Bargmann, 2012*; *Marder, 2011*). The specific signaling systems altered following deafening in this study provide a set of candidate mechanisms that may influence song. For example, corticotropin-releasing hormone binding protein (*CRHBP*) , one of the most strongly downregulated genes in RA following deafening, modulates activity in the CRH signaling pathway (*Kemp et al., 1998*), which has diverse effects on long-term potentiation, neuronal excitability, and spine dynamics in central circuits (*Aldenhoff et al., 1983*; *Blank et al., 2003*; *Chen et al., 2008*; *Fox and Gruol, 1993*; *Kratzer et al., 2013*; *Li et al., 2016*). Such evidence suggests that the dynamic modulation of neuropeptides could play a prominent role in regulating birdsong stability and plasticity and may similarly influence the control of other stable sensorimotor skills such as human speech.

Although we included the amount of singing in the two hours prior to euthanasia (a proxy for neural activity in the song system) as a variable in our regression analysis, we cannot fully dissociate the influences of motor destabilization per se and alterations to neural activity driven by the loss of auditory activity. Future work could combine similar circuit-wide gene expression analysis with disruptions to auditory input that do not alter hearing generally (such a delayed auditory feedback [*Leonardo and Konishi, 1999*]) to further characterize song plasticity-specific expression responses. Similarly, manipulations that induce song plasticity without altering hearing, such as tracheosyringeal nerve cuts (*Roy and Mooney, 2007*), may help disambiguate motor-vs-auditory expression responses. Ultimately, direct manipulations of gene expression in song regions (through knockdown or overexpression) combined with analyses of song destabilization would help clarify the causal roles of candidate genes in promoting or limiting song plasticity.

A previous study examined how the loss of auditory input before song learning in juveniles influences gene expression in HVC and RA (*Mori and Wada, 2015*). In that work, the authors identify a strong gene expression signature that varies with developmental age but is independent of whether the birds are hearing or deaf. Follow-up experiments examined the expression of a subset of developmentally-regulated genes between hearing and adult-deafened birds (similar to the approach

used here) and found no significant change. That work identified an important separation between developmentally-driven and experience-dependent molecular responses in the song system, but its aims were distinct from the present study, which sought to identify gene expression responses to deafening-induced song plasticity.

## Neuronal contributors to song plasticity

By integrating song system-wide and cell-resolved expression profiles, we can make initial predictions about which cell classes exhibit transcriptional changes during song destabilization. Glutamatergic projection neurons in RA are similar to layer 5 extratelencephalic neurons in the mammalian neocortex, both in terms of their projections to subcerebral structures and their expression profiles (*Colquitt et al., 2021*; *Nevue et al., 2020*; *Pfenning et al., 2014*; *Vicario, 1991*). A number of differentially expressed genes showed biased expression toward glutamatergic neurons in RA, including protein kinase C β (*PRKCB*), a calcium sensor associated with short-term plasticity (*Chu et al., 2014*; *Fioravante et al., 2014*) and previously shown to be upregulated in RA following deafening (*Watanabe et al., 2002*), as well as the lipid processing enzyme *LPL*, discussed in the previous section of the Discussion.

Our results also point to a prominent role for GABAergic interneurons in deafening-induced song plasticity. In particular, genes that had reduced expression with song destabilization showed an expression bias toward *Sst*- and *Pvalb*-class interneurons in RA. The interneuron subclasses present in the song system are strongly similar to well-characterized interneuron types in the mammalian neocortex, suggesting deep conservation of inhibitory networks (*Colquitt et al., 2021*). How these specific subclasses influence network activity in the song system is an open question; however, previous work has established a general role for local inhibition in the regulation of song learning and stability (*Kosche et al., 2015*; *Vallentin et al., 2016*). In particular, song learning in juveniles — during which song becomes more structured and less variable — refines the synaptic connectivity between glutamatergic projection neurons in RA and an inhibitory neuron type that has electrophysiological properties similar to fast-spiking *Pvalb*-class interneurons (*Miller et al., 2017*). Similarly, inhibitory input to HVC projection neurons increases and becomes more precise as song performance improves during juvenile song learning (*Vallentin et al., 2016*). Many of the *Sst/Pvalb*-biased genes affected by deafening-induced song plasticity are secreted neuropeptides that are sensitive to levels of neural activity (*Hou and Yu, 2013*; *Tyssowski et al., 2018*), suggesting that their reduced expression reflects reduced activity in these populations during birdsong destabilization. Moreover, several of these neuropeptides, including SST and CRHBP, act to inhibit neural activity, either directly through receptor binding or indirectly through interactions with other neuromodulators (*Hou and Yu, 2013*; *Li et al., 2016*; *Pittman and Siggins, 1981*).

This role of inhibition in maintaining birdsong structure has parallels to the role of inhibition during neural plasticity in mammals. For instance, the density of synapses from *Sst*- and *Pvalb*-class interneurons onto pyramidal neurons in the mammalian motor cortex is modulated during motor learning in mice, with an overall reduction of *Sst*-class input during motor plasticity (*Chen et al., 2015*). Likewise, low *Pvalb*-class network activity in the hippocampus is associated with increased synaptic plasticity, and low *Pvalb* expression, itself sensitive to neural activity, is found in the motor cortex during early motor learning (*Donato et al., 2013*). Similarly, increased neuronal excitability and decreased inhibition have also been found in the mammalian auditory cortex following deafening or noise trauma (*Kotak et al., 2008*; *Kotak et al., 2005*; *Seki and Eggermont, 2003*). Together, these results suggest that reduced inhibition, either through altered synaptic transmission or neuromodulation, is a key component of neural plasticity in both the mammalian and avian central nervous systems.

## Circuit contributions to transcriptional state

By sampling gene expression across the different connected components of birdsong neural circuitry in individual birds, our study allowed us to examine the correlation of gene expression in one brain area with that in another. Regions with direct projections to each other, for instance, HVC to RA and LMAN to RA, tended to have a higher number of genes with correlated expression across individuals than song or non-song regions that are not directly connected. Moreover, genes with correlated expression across the three pallial (cortical-like) song regions HVC, RA, and LMAN were enriched for activity-dependent genes and secreted neuropeptides. These results could reflect the presence

of a shared molecular state across the song system, perhaps established by singing-related neural activity or a common response to a general hormonal factor reflecting some aspect of a given bird's state (e.g. testosterone levels which may in turn be affected by sensory deprivation [*Livingston et al., 2000*]). Two results further support that these correlations reflect some aspect of shared activity across regions. First, deafened birds exhibited, on the whole, reduced gene expression correlation between song regions, suggesting that either the loss of auditory information or associated song destabilization disrupts inter-region coordination of gene expression. Second, lesions of LMAN altered gene expression specifically in RA, one of its primary efferents, but not other brain regions to which it does not directly project. This analysis highlights the importance of considering inter-regional influences in understanding the mechanistic basis of transcriptional responses across an integrated neural circuit.

The structure of the song system offers an opportunity to better understand this general issue of how connected neural components mutually influence local properties. In particular, HVC and LMAN converge onto RA and have distinct roles in song production and learning. A number of studies have examined how disrupting these inputs influences RA neural activity, synaptic transmission, neuronal morphology, and cell survival (*Akutagawa and Konishi, 1994*; *Johnson et al., 1997*; *Johnson and Bottjer, 1994*; *Kittelberger and Mooney, 1999*; *Ölveczky et al., 2011*), yet little is known about how each afferent differentially influences the molecular features of RA. A joint analysis of how each afferent alters gene expression across diverse RA neuronal types, using circuit-wide and cell-resolved gene expression approaches such as those described here, could yield insight into how converging neural inputs are integrated in target structures at the molecular level. Our current analysis focused on the transcriptional effects of unilateral LMAN lesions on target structures in hearing birds. We found that expression differences in RA following LMAN lesions were broadly the inverse of those following deafening, suggesting that the loss of LMAN establishes a transcriptional state characteristic of reduced plasticity. An informative followup experiment would be to perform bilateral lesions of LMAN before cochlear removal — a manipulation known to prevent deafening-induced song destabilization (*Brainard and Doupe, 2000*; *Kojima et al., 2013*; *Scott et al., 2000*) — and compare expression profiles in RA in these birds to those in birds with only bilateral LMAN lesions, only cochlear removal, or unmanipulated controls. We predict that this approach would uncover genes whose expression tracks with song destabilization across manipulations, further pinpointing relevant plasticity-associated molecular factors.

These cross-regional patterns of gene expression relate to a central question of this study: how does the loss of sensory input influence gene expression in sensorimotor circuits? Our results suggest a model in which altered activity propagates through existing circuits, such that the state of one circuit component progressively modifies gene expression in its targets. Local mechanisms engaged within each region, such as synapse/spine remodeling and neuropeptidergic signaling implicated here, could then alter circuit connectivity and function, leading to behavioral plasticity. Identifying how neural activity influences gene expression across neural circuits and what specific molecular and cellular factors in turn shape circuit function will be instrumental to better understand the neural mechanisms that underlie sensorimotor stability and its impairment following sensory loss.

## Materials and methods

**Key resources table**

| Reagent type (species) or resource | Designation | Source or reference | Identifiers | Additional information |
|---|---|---|---|---|
| Biological sample (*Lonchura striata domestica*) | Brain tissue | Lab animal colony | | |
| Sequence-based reagent | RT_primer_v1 | IDT | SLCR-seq primer | AAGCAGTGGTATCAACGCAGAGTA CNNNNNNNNNNNNNNNNNNNNNNN NNXXXXXXTTTTTTTTTTTTTTTTTTTT TTTTTTTTTVN |
| Sequence-based reagent | RT_primer_v2 | IDT | SLCR-seq primer | AAGCAGTGGTATCAACGCAGAGTA CNNNNNNNNNNNNNNNNATCTAGCCGG CCTTTTTTTTTTTTTTTTTTTTTTTTTTTVN |

*Continued on next page*

*Continued*

| Reagent type (species) or resource | Designation | Source or reference | Identifiers | Additional information |
|---|---|---|---|---|
| Sequence-based reagent | TSO_LNA | Exiqon | SLCR-seq primer | AAGCAGTGGTATCAACGCAG AGTGAATrGrG +G |
| Sequence-based reagent | TSO_PCR | IDT | SLCR-seq primer | AAGCAGTGGTATCAACGCAGAGT |
| Sequence-based reagent | P5-TSO_Hybrid | IDT | SLCR-seq primer | AATGATACGGCGACCACCGAGAT CTACACGCCTGTCCGCGGAAGCA GTGGTATCAACGCAGAGT*A*C |
| Sequence-based reagent | Read1CustomSeqB | IDT | SLCR-seq primer | GCCTGTCCGCGGAAGCAGTG GTATCAACGCAGAGTAC |
| Sequence-based reagent | PCR2 | IDT | SLCR-seq primer | CAAGCAGAAGACGGCATACGAGA TYYYYYYYYYGTCTCGTGGGCTCGG |
| Commercial assay or kit | KAPA HiFi Hotstart | Roche | 7958927001 | |
| Commercial assay or kit | Qubit dsDNA HS | ThermoFisher | Q32851 | |
| Commercial assay or kit | Nextera XT | Illumina | FC-131–1024 | |
| Commercial assay or kit | KAPA Library Quantification Kit | Roche | 07960140001 | |
| Commercial assay or kit | 2% BluePippin Gels | Sage | BEF2010 | |
| Commercial assay or kit | ISH-HCR | Molecular instruments | ISH probes and reagents | |
| Commercial assay or kit | AMPure XP | Beckman Coulter | A63881 | |
| Other | molecule sieve beads | Sigma | 208582 | Used to make anhydrous ethanol solution used in 'SLCR-seq — rapid Nissl stain' |
| Other | cresyl violet powder | Sigma | 255246 | Stain used in 'SLCR-seq — rapid Nissl stain' |
| Other | Guanidine thiocyanate | Sigma | G9277 | Component of lysis solution used for RNA purification in 'SLCR-seq — SPRI RNA purification' |
| Other | Sera-Mag SpeedBeads Carboxyl Magnetic Beads, hydrophobic | FisherScientific | 09-981-123 | Used to create homemade SPRI RNA purification solution, as in 'SLCR-seq — SPRI RNA purification' |
| Other | EvaGreen | Biotium | 13000 | Dye added to library amplification to determine needed number of amplification cycles, as in 'SLCR-seq — library preparation' |

## Animal care and use

All Bengalese finches were from our breeding colonies at UCSF or were purchased from approved vendors. Experiments were conducted in accordance with NIH and UCSF policies governing animal use and welfare.

## Song recording and preprocessing

Birds were individually housed in wire cages in sound isolation chambers. Song was recorded using Countryman Isomax microphones taped to the top of the wire cage. Microphones were connected to USB preamplifiers that were connected to a Linux workstation. Audio was recorded at a frame rate of 44,100 samples/second using a custom python script, and, to select for periods of singing,

blocks of continuous sound with amplitudes above a manually set threshold were saved as 24-bit WAV files.

## Song autolabeling

The analysis of specific spectral features (e.g. fundamental frequency) was performed on syllables that were labeled using a supervised machine learning approach, called hybrid-vocal-classifier or *hvc* (*Nicholson, 2021*). For each bird, 20–50 songs were manually labeled using the Matlab software evsonganaly. Using *hvc*, a set of spectrotemporal features was computed for each syllable (e.g. duration, mean frequency, pitch goodness, and mean spectral flatness as defined in *Tachibana et al., 2014*). These features and the manually defined labels were provided to *hvc* to train a support vector machine (SVM) with radial basis function, with a grid search across parameters C and gamma to identify parameters with the highest classification accuracy. A set of models were then trained using these selected parameters and a random sample of training syllables, and model accuracy was tested on a held-out set of syllables. For each bird, the number of input syllables and parameters were adjusted until label accuracy reached 95–100%. The model with the highest accuracy was then used to predict labels on unlabeled songs. To select confidently labeled syllables, a prediction confidence score was calculated for each syllable as the entropy (sklearn.stats.entropy) of the classification probabilities resulting from SVM model prediction. Syllables with a prediction confidence score greater than 0.5 were retained.

## Song dimensionality reduction

To project syllable spectrotemporal structure into a reduced dimension space, we used an approach developed by *Sainburg et al., 2020c* with code and example scripts obtained from the AVGN Github repository (*Sainburg, 2020b*). Songs were first isolated from audio recordings and then segmented into syllables based on amplitude threshold crossings. Spectrograms were computed for each syllable using short-time Fourier transforms (512 window size, 0.5 ms step size, 6 ms window size, 44,100 frames per second) and frequencies between 500 Hz and 15,000 Hz were retained. Spectrograms were converted to mel scale using a mel filter with 128 channels. Syllables were compressed in the time dimension to a framerate of 640 frames per second then zero-padded to yield a standardized dimension of 128. Before dimensionality reduction, these 128 × 128 spectrograms were further reduced to 16 × 16 matrices and then flattened yielding a 256-length feature vector for each syllable. Syllable x feature vector matrices were then processed using the single-cell analysis framework *Seurat v3* (*Stuart et al., 2019*). Principal component analysis was performed, then Uniform Manifold Approximation and Projection (UMAP) was performed on the first 10 principal components to produce a two-dimensional reduction.

## UMAP density differences

To calculate global differences in syllable spectral structure before and after a manipulation (e.g. deafening), we split each bird's song UMAP by day relative to the manipulation and computed two-dimensional kernel density estimates (R package *MASS* v7.3 function *kde2d*, 200 × 200 grid) for each of these per-day plots. A baseline UMAP structure was calculated as the mean density across the 2–4 days of singing before the manipulation, then density differences were calculated by subtracting this baseline density from each per-day density plot. Positive values from each difference plot were summed to give a single statistic for each day. Significance between hearing and deaf conditions for each day was determined using a two-sided *t*-test.

## Fundamental frequency statistics

To calculate fundamental frequency (FF) for a given harmonic stack, we first computed the average spectrogram for 20 randomly selected syllables. We then identified a time within the syllable (relative to syllable onset) with stable FF and defined minimum and maximum frequency bounds to define a frequency band containing the FF. A short-time Fourier transform (STFT) was then calculated at this time point using function *spec* from R package *seewave* v2.1.8 (*Sueur et al., 2008*) (1024 window size, 44,100 frames per second). FF was estimated by interpolating the frequency spectrum on an output vector spanning the minimum and maximum frequency bounds with a resolution of 1 Hz (function *aspline* from R package *akima* v0.6–2.2). The maximum value of this interpolated frequency spectrum

was taken as the FF. Rolling variability of FF was calculated as the coefficient of variation (CV, standard deviation/mean) over a set of FF values for a given syllable and the 10 prior syllables (of the same type). To compare variability relative to a baseline period, FF CV values were transformed to a percentage relative to the average variability before manipulation. Group estimates and significance values were obtained from mixed effects linear models using R package *lme4* v1.1–27.1 (*Bates et al., 2015*) and function *lmer* (maximum likelihood criterion). The time period (before or after manipulation) was treated as a fixed effect and bird ID and syllable were treated as random effects with syllable nested under bird ID [model in *lme4* notation: period + (1 | bird/syllable)]. p-values for fixed effect were obtained using ANOVA (Type II, Wald chi-square test statistic, R package *car* v3.0–11 function *Anova, Fox and Weisberg, 2019*) followed by adjustments for multiple testing using Benjamini-Hochberg correction.

## Song $D_{KL}$

To provide a single statistic that represents the amount of difference between songs in two conditions, we used a measure that we previously developed called Song $D_{KL}$ (*Mets and Brainard, 2018*). Songs for a given bird were divided into 'pre' and 'post'-procedure groups. The 'pre' group consisted of songs from at most four days before the procedure up to the day preceding the procedure. The 'post' group contained song from two days before the day of euthanasia to the day of euthanasia. A maximum of 50 songs were sampled from each day. Syllables were identified in song WAV files by amplitude thresholding using a manually defined threshold for each bird. Mean power spectral densities (PSD) were computed for each segmented syllable using short-time Fourier transforms via R package seewave v2.1.8 (*Sueur et al., 2008*) and function meanspec (window length 'wl'=512, overlap 'ovlp'=0%, normalized 'norm'=T). Syllables in each dataset were split into a training dataset of 500 syllables and a held-out dataset of the remaining syllables. 50 PSDs were randomly selected from the 'pre' training dataset to serve as reference syllables for distance calculations. Inter-syllable spectral distances were calculated as Euclidean distances between this reference syllable set and each PSD, generating distance matrices for the 'pre' training and held-out datasets and the 'post' training and held-out datasets. Gaussian mixture models (GMMs) were fit to the 'pre' training distance matrix using function Mclust (5–12 mixture components, diagonal multivariate mixture model with varying volume, varying shape 'VVI') from R package mclust v5.4.7 (*Scrucca et al., 2016*). Bayesian Information Criterion (BIC) was computed for each model and second-order differences (difference of the difference) were calculated between the BICs for models with increasing numbers of mixture components. The model with minimum second-order BIC difference was selected for further use. A GMM was likewise fit to the 'post' training distance matrix using the same number of mixture components as in the selected 'pre' training model. The likelihood of generating each syllable in the 'pre' held-out dataset under the 'pre' and 'post' GMMs was calculated. This procedure was repeated ten times with different randomly selected reference syllables. The Kullback-Leibler divergence was then calculated as

$$D_{KL} = log_2 \left( \overline{L_1} - \overline{L_2} \right)$$

where $\overline{L_1}$ is the mean likelihood of observing a 'pre' held-out syllable across the ten replicated 'pre' models and $\overline{L_2}$ is the corresponding mean likelihood value for the 'post' models. These syllable-level $D_{KL}$ values were then averaged to give a single $SongD_{KL}$ for a given bird.

## Deafening by bilateral cochlear removal

Nine Adult male Bengalese finches (103–458 days post-hatch, median ± SD of 133 ± 123) were deafened by bilateral cochlear removal. Birds were anesthetized using isoflurane and an incision was made in the skin covering the ear canal to expose the canal. The tympanic membrane was ruptured, and the columella was removed using forceps. Cochlea were removed using a fine tungsten wire shaped into a hook. The incision was then resealed using VetBond (3M). For each deafened bird, a control ('hearing') bird underwent a sham surgery on the same day in which the bird was anesthetized, and the skin incision was made and then resealed. Birds survived for 4, 9, or 14 days (three hearing and three deaf birds for each timepoint) then were euthanized as described in *Euthanasia and brain preparation*.

## Unilateral LMAN lesions

Five birds received unilateral LMAN lesions, three with left-hemisphere lesions, and two with right-hemisphere lesions. LMAN was electrolytically lesioned using a 100 kOhm platinum/iridium electrode.

LMAN was stereotactically located at 4.7 mm AP, 1.7 mm ML, and 2.1 mm DV using a beak angle of 50 degrees. In one hemisphere, five penetrations were made, one at the given coordinates and four more +/-300 microns from this center position. At each penetration, 100 μA of current at the anode was passed for 60 s. At the experiment end, birds were euthanized as described in *Euthanasia and brain preparation*. To assess lesion completeness, 20 um coronal cryosections were collected at 100 um intervals across the anterior-posterior extent of LMAN onto SuperFrost Plus slides (Fisher-Brand), then Nissl stained as described in *Standard Nissl stain, fresh-frozen cryosections*. The volume of LMAN was estimated in ImageJ/Fiji (*Schindelin et al., 2012*) by calculating the area of LMAN on the unlesioned side in each section that it was visible, interpolating a smooth curve in R across these measured areas and the known distance between cryosections, then calculating the area under the curve. This procedure was repeated for any residual LMAN visible on the lesioned side, and the lesion percentage was calculated as 100 * (1 - [volume LMAN lesioned] / [volume LMAN unlesioned]). LMAN was considered lesioned if more than 25% of the volume was spanned by the lesion.

### Standard Nissl stain, fresh-frozen cryosections

Frozen sections were allowed to come to room temperature for at least 20 min and then placed in a glass staining rack. Then slides were sequentially transferred to two rounds of xylenes for 5 min, two rounds of 100% ethanol for 5 min, one round of 95% ethanol for 5 min, 1 round of 70% ethanol for 5 min, water for 1 min, stained in 0.5% cresyl violet solution for 30 min, then rinsed for 1 min in water. Slides were then transferred to one round of 70% ethanol for 15–20 s (depending on desired staining intensity), one round of 95% ethanol for 30 s, two rounds of 100% ethanol for 30 s each, then two rounds of xylenes for 3 min each. DPX Mountant (Sigma) was applied, then slides were coverslipped. 0.5% cresyl violet was prepared as 300 mL water, 1 mL glacial acetic acid, and 1.5 g cresyl violet acetate. Solution was stirred for two days with no heat and then filtered.

### Euthanasia and brain preparation

Birds were euthanized using isoflurane, decapitated, and debrained. All birds used for Serial Laser Capture RNA-seq were euthanized 2 hr after lights on at 9 AM. Brains were flash-frozen in –70 C dry ice-chilled isopentane for 12 s within 4 min from decapitation.

### Serial laser capture microdissection RNA-sequencing (SLCR-seq) — overview

We were motivated by improvements to low-input RNA-sequencing stemming from optimized single-cell approaches to develop a method that would allow the construction of tens to hundreds of gene expression libraries from anatomically-defined regions. To achieve this we combined an optimized rapid Nissl staining protocol, laser capture microdissection, scalable RNA purification, and low-cost and low-input RNA-sequencing library construction into a single pipeline called Serial Laser Capture Microdissection RNA-sequencing (SLCR-seq).

### SLCR-seq — cryosectioning

Surfaces in the cryostat chamber were first cleaned using a mixture of 50% RNaseZap (Ambion)/50% ethanol followed by a rinse of 70% ethanol in nuclease-free water. Flash-frozen brains were removed from –80 °C storage and allowed to equilibrate in a cryostat chamber set to –18 °C for ~30 min. PEN membrane slides for LCM (Leica) and Superfrost Plus glass slides for histology (Fisherbrand) were placed in the cryochamber to chill. Once equilibrated, the brain was mounted onto a cryostat chuck using a small amount of OCT (TissueTek) with the posterior surface down and the anterior surface available for coronal sectioning. The brain was trimmed approximately 1.8 mm until reaching the anterior-posterior position of LMAN and Area X, which were visible as slightly darker regions. Sections were cut at 20 μm, transferred to pre-chilled membrane or glass slides, then melted onto the slides using a metal dowel that was pre-warmed on a slide warmer. Once a section was fully melted, the slide was transferred to a metal block in the cryostat chamber to refreeze the section. After sectioning through LMAN and Area X, the brain was detached from the chuck and remounted along the cut anterior surface for sectioning from the posterior surface. The brain was trimmed until reaching the anterior-position for RA (~0.8 mm from the posterior surface of the forebrain), which was also evident as a slightly darker region. Sections were collected onto membrane and glass slides as

described through the level of HVC (~2.3 mm from the posterior surface) or Field L (visible as a dark curve extending from the medial surface). Once the collection was finished, slides were transferred to plastic slide mailers and stored in freezer boxes at –80 °C. Remaining brain tissue was re-wrapped in aluminum foil, placed back into a 15 mL conical tube, and stored at –80 °C. Test assays indicated that brains could be resectioned once more (for a total of two sectioning sessions) without negatively impacting RNA quality.

## SLCR-seq — rapid Nissl stain

A fast Nissl staining procedure was developed to quickly stain cryosections before laser capture microdissection. Anhydrous 100% ethanol solution was prepared by adding 15 g of molecular sieve beads (Sigma 208582, 3 Å, 8–12 mesh) to 500 mL 100% molecular grade ethanol (Sigma E7023). Cresyl violet staining solution was prepared as by dissolving cresyl violet powder (Sigma) to 4% wt/vol in 75% ethanol (75% molecular grade ethanol, 25% nuclease-free water), stirring for two days, then filtering through a 0.22 μm filter. Before staining, a series of 95%, 75%, and 50% ethanol solutions were prepared. To stain, each slide was thawed at room temperature on a bench for 20 s then transferred to 95% ethanol for 30 s, 75% ethanol for 30 s, and 50% ethanol for 30 s. 400 μL of cresyl violet staining solution was then applied to the slide for 30 s. Slides were destained and dehydrated by transferring to 50% ethanol for 30 s, 75% ethanol for 30 s, 95% ethanol for 30 s, then two rounds of 100% ethanol for 30 s each. Slides were then allowed to air dry. Time series experiments indicated that RNA quality was maintained for up to 45 min following staining.

## SLCR-seq — Laser capture microdissection

After staining, slides were loaded onto a Leica LMD7000. Song nuclei were identified by anatomical landmarks (such as lamina and position relative to brain surfaces) and their higher intensity Nissl staining relative to surrounding regions. Sections cut from the surrounding tissue (power 45, aperture 50, speed 10, specimen balance 0, head 90%, pulse 92, offset 15) into eight-well strip caps containing 31.5 μL of RNA Lysis Buffer/PK (see *SPRI RNA purification*). After filling each cap with a section, the strip was placed onto a 96-well plate pre-chilled on ice and covered with an ice pack. Once a plate was filled, it was vortexed, spun down at 3250 × g for 5 min at 4 °C, then transferred to dry ice. For long-term storage, plates were stored at –80 °C.

## SLCR-seq — SPRI RNA purification

The following solutions were prepared before LCM section collection: 50% guanidine thiocyanate (Sigma) in nuclease-free water, 5 X CN buffer (250 mM sodium citrate pH 7.0 (Sigma), 5% NP-40 (Sigma)), and RNA Lysis Buffer (20% guanidine thiocyanate, 1 X CN buffer). The following solutions were prepared before RNA purification: RNA Wash Buffer (25 mM sodium citrate pH 7.0, 15% guanidine thiocyanate, 40% isopropanol) and solid phase reversible immobilization (SPRI) bead solution. SPRI bead solution was prepared by first vortexing Sera-Mag SpeedBeads Carboxyl Magnetic Beads, hydrophobic (Fisher) until fully suspended transferring 1 mL beads to a 1.5 mL tube. Beads were washed by placing the tube on a tube magnet, waiting until the solution cleared, removing the solution, adding 1 mL of TE Buffer (10 mM UltraPure Tris HCl, pH 8.0 (ThermoFisher), 1 mM EDTA pH 8 (ThermoFisher)), and pipetting to mix. This wash was repeated once more, then the beads were resuspended in 1 mL TE Buffer. Separately, 9 g polyethylene glycol 8000 (Amresco), 10 mL 5 M NaCl, 500 μL 1 M UltraPure Tris HCl pH 8.0 (ThermoFisher), 100 μL 0.5 M EDTA pH 8.0 (ThermoFisher), and 500 μL 2% sodium azide (Sigma) were combined and brought to ~49 mL using nuclease-free water. Solution was mixed by inversion until PEG 8000 went into solution. Then, 137.5 μL of 20% Tween-20 and 1 mL of beads/TE were added and mixed by inversion. This SPRI bead solution was then stored at 4 °C.

Just before LCM collection, 31.5 μL of RNA Lysis Buffer/PK (1.5 μL of Proteinase K (Ambion), 30 μL of RNA Lysis Buffer) was prepared for each well. To purify RNA following LCM section collection, samples were first allowed to thaw on ice if stored at –80 °C. SPRI bead solution was allowed to come to room temperature, then 40 uL SPRI bead solution was mixed with 47.5 uL isopropanol for each sample. Samples were then lysed by incubating at 42 °C for 30 min in a thermocycler and then placed at room temperature. 87.5 uL of SPRI/isopropanol solution was added to each sample and then mixed 10 x by pipetting. Samples were incubated for 5 min at room temperature and then transferred to

a magnetic plate stand. After 3 min, the solution was removed, the plate was removed from the magnetic stand, 100 uL of RNA Wash Buffer was added, and beads were resuspended by pipetting. The plate was immediately transferred back to the magnetic plate stand and held there for 2 min until the solution cleared. The solution was removed, and the plate was removed from the stand. 100 uL of 70% ethanol was added, beads were resuspended by pipetting 10 times, the plate was returned to the magnetic stand, the solution was allowed to clear for 2 min, and the solution was removed. This step was repeated for two total ethanol washes. Following the final wash, the beads were allowed to dry for 10 min while the plate remained on the stand. Residual ethanol was removed by pipetting. To elute RNA, the plate was removed from the magnet, 15 uL of nuclease-free water was added to each sample, and beads were resuspended by pipetting 10 times. Samples were incubated at room temperature for 5 min, then the plate was transferred to a low-elution volume magnetic stand. After 2 min or until the solution cleared, 10–12 µL eluted RNA was transferred to new 96-well plates on ice. Plates were sealed using foil adhesive, frozen on dry ice, then transferred to –80 °C for long-term storage.

## SLCR-seq — library preparation

The SLCR-seq library preparation was adapted from several low-input and single-cell RNA-sequencing library protocols (*Islam et al., 2014*; *Islam et al., 2012*; *Kivioja et al., 2012*; *Macosko et al., 2015*; *Picelli et al., 2014*). Barcoded unique molecular identifier (UMI) reverse transcription (RT) primers were prepared in advance in a 96-well plate (RT/TSO/dNTP mix). Each well contained 10 µM barcoded reverse transcription primer (RT_primer, IDT), 10 µM template-switching oligonucleotide with lock nucleic acids (TSO_LNA, Exiqon), and 10 mM dNTPs. Plates were sealed with foil adhesive and stored at –80 °C. Two RT primers were used in this study: one for the initial 18 bird deafening dataset (RT_primer_v1, 25 base UMI, six base barcode), and another for the 10 bird unilateral LMAN dataset (RT_primer_v2, 14 base UMI, 12 base barcode). RT_primer_v1 and RT_primer_v2 sets consisted of 24 and 48 barcodes, respectively (*Supplementary file 1*). Barcodes were at least one edit distance away from all other barcodes in the set.

For library preparation, total RNA prepared from *SPRI RNA purification* was thawed on ice, then 4 µL total RNA was placed into a well of a 96-well plate chilled on ice. 1 µL RT/TSO/dNTP mix was added and mixed 10 times by pipetting. Plates were sealed with foil adhesive, incubated at 72 °C for 3 min, then snap-cooled in ice for at least 2 min. An RT Master Mix was prepared containing 1 x Enzscript RT buffer (Enzymatics), 5 mM dithiothreitol, 1 mM betaine, 12 mM $MgCl_2$, 0.25 µL Recombinant Ribonuclease Inhibitor (Takara), and 10 U/µL Enzscript Moloney-Murine Leukemia Virus Reverse Transcriptase (Enzymatics). 5 µL of RT Master Mix was added to each sample and mixed by pipetting 10 times. Plates were sealed with foil adhesive and incubated in a thermocycler: 42 °C for 90 min, 70 °C for 15 min, 4 °C hold. Reactions were then pooled within a barcode set (e.g. barcodes 1–48 from RT_primer_v2 were combined into one tube). To purify cDNA, 0.6 x volume of Ampure XP bead solution was added to each pooled sample and mixed by pipetting 10 times. Samples were incubated for 5 min and transferred to a tube magnet. After the beads cleared from the solution, the solution was removed, and the beads were washed in 400 µL freshly prepared 80% ethanol for 30 s. This step was repeated for a total of two washes. After the second wash, the ethanol solution was removed, and the beads were allowed to dry for 5–10 min. Beads were then resuspended in 22 µL of nuclease-free water and incubated for 2 min. 20 µL eluted cDNA was transferred to new 1.5 mL LoBind tubes or 96-well plates and either stored at –20 °C or amplified immediately.

During the purification a 40 µL cDNA Amplification Master Mix was prepared containing 10 µL KAPA HiFi 5 x Buffer, 1 µL 10 mM dNTPs, 4 µL 10 mM TSO_PCR primer, 0.5 µL 1 U/µL KAPA HiFi Hotstart DNA polymerase, and 24.5 µL nuclease-free water. 10 µL of purified cDNA was added to this master mix, pipetted 10 x to mix, then amplified under the following cycling parameters: 95 °C for min, then four cycles of 98 °C for 30 s, 65 °C for 45 s, and 72 °C for 3 min. Reactions were then placed on ice. During this initial amplification, a second master mix was prepared to determine the target number of amplification cycles by quantitative PCR. This mix contained 3 µL KAPA HiFi 5 x Buffer, 0.3 µL 10 mM dNTPs, 1.2 µL 10 mM TSO_PCR primer, 0.15 µL 1 U/µL KAPA HiFi Hotstart DNA polymerase, 0.75 µL 20 x EvaGreen (Biotium), and 4.6 µL nuclease-free water. 5 µL of preamplified cDNA was added to this mix and amplified in a real-time PCR machine: 98 °C for 3 min, followed by 24 cycles of 98 °C for 20 s, 67 °C for 20 s, and 72 °C for 3 min, followed by 72 °C for 5 min. The target number

of additional cycles was determined by identifying the Ct at 20% of the max fluorescence and then subtracting five cycles from this number. This number was generally between 5–7 additional cycles. The remaining 45 µL was placed back into the thermocycler and cycled at 98 °C for 30 s, the number of additional cycles at 98 °C for 20 s, 67 °C for 20 s, and 72 °C for 3 min, followed by 72 °C for 5 min.

To purify the amplified cDNA, 0.6 x volume of Ampure XP bead solution was added to each reaction and mixed by pipetting 10 times. Samples were incubated for 5 min and transferred to a tube magnet. After the beads cleared from the solution, the solution was removed, and the beads were washed in 200 µL freshly prepared 80% ethanol for 30 s. This step was repeated for a total of two washes. After the second wash, the ethanol solution was removed, and the beads were allowed to dry for 5 min. Beads were then resuspended in 22 µL of nuclease-free water and incubated for 2 min. 20 µL eluted cDNA was transferred to new 1.5 mL LoBind tubes or 96-well plates and stored at –20 °C. Sample concentration was quantified using Qubit dsDNA High Sensitivity kit (ThermoFisher), then sample concentrations were standardized to 100 pg/µL.

To prepare tagmented DNA, 4 µL (400 pg) of amplified cDNA was added to 10 µL Tagmentation Buffer (Buffer TD from the Nextera XT DNA Sample Prep Kit, Illumina), 1 µL nuclease-free water, and 5 µL ATM (Nextera XT). Reactions were mixed by pipetting 10 times the incubated at 55 °C for 5 min. 5 µL of Buffer NT was then added, then the reactions were incubated for 5 min at room temperature.

Final libraries were constructed by first preparing a PCR master mix containing 20 µL KAPA HiFi 5 x Buffer, 2 µL 10 mM dNTPs, 5 µL 10 mM P5-TSO_Hybrid primer, 5 µL 10 mM PCR2 primer, 1 µL 1 U/µL KAPA HiFi Hotstart DNA polymerase, and 42 µL nuclease-free water. PCR2 contains an i7 index (*Supplementary file 1*). The 25 µL tagmentation reaction was then added directly to the mix, and mixed by pipetting 10 times. Samples were amplified using 72 °C for 3 min; 95 °C for 3 min; followed by 16 cycles of 98 °C for 10 s, 55 °C for 30 s, and 72 °C for 30 s; followed by 72 °C for 5 min. Samples were then purified by adding 1.2 x volumes of Ampure XP, incubating for 5 min, then transferring to a tube magnet. After the beads cleared from the solution, the solution was removed, and the beads were washed in 200 µL freshly prepared 80% ethanol for 30 s. This step was repeated for a total of two washes. After the second wash, the ethanol solution was removed, and the beads were allowed to dry for 5 min. Beads were then resuspended in 22 µL of Low Elution Buffer (10 mM Tris HCl pH 8.0, 0.1 mM EDTA, 0.05% Tween-20) and incubated for 2 min. 20 µL eluted cDNA was transferred to new 1.5 mL LoBind tubes and stored at –20 °C. Library size distributions were assessed using a Bioanalyzer High Sensitivity DNA Chip (Agilent), and library concentrations were determined using the KAPA Library Quantification Kit (Illumina Complete Kit, Roche). Samples were pooled at equal concentrations and then size-selected using a BluePippin and 2% BluePippin gels. DNA from 180 to 500 bp was selected and then purified using the MinElute kit (Qiagen) with two rounds of 10 µL elution in Low Elution Buffer. Samples were stored at –20 °C.

| | |
|---|---|
| RT_primer_v1 | AAGCAGTGGTATCAACGCAGAGTACNNNNNNNNNNNNNNNNNNNNNNNNN XXXXXXTTTTTTTTTTTTTTTTTTTTTTTTTTTTTTTTTVN |
| RT_primer_v2 | AAGCAGTGGTATCAACGCAGAGTACNNNNNNNNNNNNNNNNATCTAGCCGG CCTTTTTTTTTTTTTTTTTTTTTTTTTTTTTTTVN |
| TSO_LNA | AAGCAGTGGTATCAACGCAGAGTGAATrGrG +G |
| TSO_PCR | AAGCAGTGGTATCAACGCAGAGT |
| P5-TSO_Hybrid | AATGATACGGCGACCACCGAGATCTACACGCCTGTCCGCGGAAGCAGT GGTATCAACGCAGAGT*A*C |
| Read1CustomSeqB | GCCTGTCCGCGGAAGCAGTGGTATCAACGCAGAGTAC |
| PCR2 | CAAGCAGAAGACGGCATACGAGATYYYYYYYYYGTCTCGTGGGCTCGG |

'N', random nucleotide; 'X', barcode sequence; 'Y', i7 index sequence 'V', A or C or G; 'r', ribonucleic acid; '+', locked nucleic acid; *, phosphorothioate

## RNA-sequencing preprocessing

Sequencing reads were first trimmed for adaptor sequences using trim_galore (*Krueger, 2020*, `--quality` 20, `--paired`, `--overlap` 10, adaptors AAAAAAAAAA and GTACTCTGCGTTGATA CCACTGCTTCCGCGGACAGGCGTGTAGATCT). We first generated an initial alignment to the Bengalese finch genome (lonStrDom2, GCF_005870125.1) using *STAR* v2.7.8a (STARsolo mode,

default parameters, --outFilterIntronMotifs RemoveNoncanonical) (*Dobin et al., 2013*). To better annotate the 3' UTRs of Bengalese finch gene models, we identified transcript 3' ends by assembling transcripts using these initial alignments and the RNA-seq assembler *Stringtie* (*Kovaka et al., 2019*) (--fr -m 100). These Stringtie models were then intersected with the NCBI Bengalese finch transcriptome (lonStrDom2, GCF_005870125.1). New Stringtie exons were filtered by same-strandedness to the intersected reference genes, a minimal expression level (at least 10% of expression max for a given gene), and at least within 10 kilobases from the 3' end of the gene. 3' UTRs of the reference transcriptome were extended out to these new exons. Reads were then re-aligned to this extended transcriptome using the 'bus' subcommand from *kallisto* v0.46.1 *Bray et al., 2016*; *Melsted et al., 2021* followed by barcode error correction using *bustools* v0.39.3 'correct,' sorting using 'sort,' and read counting using 'count.'.

## Differential expression analysis

Gene-sample count matrices were filtered to remove lowly expressed genes, defined as having a total number of reads across samples less than the number of samples divided by eight (the number of brain regions assayed). For each sample we also calculated the 'cellular detection rate (CDR)' or the number of genes detected in a given sample, previously shown to substantially influence differential expression analysis on single-cell RNA-sequencing samples (*Finak et al., 2015*). Low-quality samples were defined as having a CDR of less than 30% of the total number of genes in the reference annotation (18,674 genes). Normalization factors were calculated using the function *calcNormFactors* from the R package *edgeR* v3.31.4 and the 'TMMwsp' method. The count matrix, these normalization factors, and a design matrix were then provided to the function *voom* from the *limma* package v.3.48.3 (*Law et al., 2014*; *Ritchie et al., 2015*). The design matrix was specified as:

~0 + position + position:num_songs_on_euth_date_log_scale + position:kl_mean_log_scale_cut2_proc2 + position:kl_mean_log_scale_cut2_proc2:num_songs_on_euth_date_log_scale + position:nsongs_per_day_pre_log_scale + cdr_scale + frac_mito_scale + sv1 + sv2

where 'position' is an indicator for brain region, 'num_songs_on_euth_date_log_scale' is log-transformed total number of songs sung on the day of euthanasia, 'kl_mean_log_scale_cut2_proc2' is log-transformed Song $D_{KL}$ discretized into three equally sized bins, 'nsongs_per_day_pre_log_scale' is log-transformed average number of songs sung per day during the pre-procedure period, 'cdr_scale' is CDR, and 'frac_mito' is the fraction of reads mapping to mitochondrial genes in a given sample. Variables with 'scale' in their names were mean-subtracted and standard deviation-normalized. 'sv1' and 'sv2' correspond to the top two surrogate variables calculated using the function *svaseq* from the R package *sva* v3.40.0 (*Leek, 2014*), with full model specified as above and a null model given as '~0 + position +cdr_scaled +frac_mito_scale.' Because SLCR-seq samples taken from the same bird and brain region are not fully independent samples, we considered these samples as technical replicates. We used the consensus correlation approach implemented in limma/voom to estimate the within-block (bird/region) expression similarity. To calculate the within-block correlation between samples, the resulting voom object was passed to *duplicateCorrelation* with block specified as the bird ID and brain region (for the deafening samples) or bird ID and brain hemisphere (for the unilateral LMAN lesion samples). To fit the model, the voom object, design matrix, and the consensus correlation were input to function *lmFit* from limma. Coefficient estimates and standard errors for each coefficient were calculated using function *contrasts.fit*, the function *eBayes* was used to compute moderated t-statistics and p-values, and the function *topTable* was used to adjust p-values using the Benjamini-Hochberg method. Genes were considered differentially expressed if their adjusted p-values were less than 0.1.

Differentially expressed genes in the unilateral LMAN lesion SLCR-seq dataset were calculated similarly but with design specified as:

~0 + group + tags + cdr_scale

where group indicates whether the region is ipsilateral or contralateral to the LMAN lesion. The variable 'tags' refers to bird ID tags and, therefore, controls for bird-level differences allowing pairwise comparisons of the effect of lesioning within birds.

Expression estimates and standard errors for a given bird and brain region were computed using a regression approach with a design matrix specified as:

~0 + position:tags + cdr_scale +frac_mito_scale

where 'position' is an indicator for brain region, 'tags' is the bird ID, 'index2' is a categorical variable indicating the sequencing run, and 'cdr_scale' and 'frac_mito_scale' are as described above. Standard errors were extracted from the linear fit model 'fit' as: sqrt(fit$s2.post) * fit$stdev.unscaled.

## Network analysis

We used the R package *MEGENA* (Multiscale Clustering of Geometrical Network, v1.3.7) to identify modules of genes with correlated expression across SLCR-seq data (*Song and Zhang, 2015*). Low-quality samples were removed by retaining samples with cellular detection rates above 0.42. Samples expression values were normalized using normalization factors calculated as described in 'Differential expression analysis'(*calcNormFactors* and the 'TMMwsp' method) and then log-transformed with a pseudocount of 1. Samples were then split by brain region. To remove batch effects contributed by which pool a given sample was in, we used the function *ComBat* (*Johnson et al., 2007*) from the R package *sva* v3.40.0. For each brain region, we then selected the top 2000 variable genes as defined using the *Seurat* function *FindVariableFeatures* v4.0.4 and the 'vst' method. Signed Pearson correlations between every pair of genes were then calculated using the function *calculate.correlation* from *MEGENA,* which calculates false discovery rates by permutation (50 permutations). Correlations with an FDR less than 0.05 were retained. We passed these pairwise correlations to function *calculate.PFN* to generate a more sparse network that retains information edges using the MEGENA Planar Filtered Network algorithm. Module detection was then performed on this filtered network using the function *do.MEGENA*.

To identify modules associated with behavioral features, we calculated the average of log-transformed estimates for a given coefficient across genes in a given module. To identify modules with greater (or lesser) than expected fold-changes, for each module we randomly selected the same number of genes and averaged their log-transformed coefficient estimates 100 times. Modules that had averages less than 1% or greater than 99% of this null distribution were considered significant. Hub genes were designated using the approach defined in MEGENA. For each module, the link weights of the planar filtered network were permuted 100 times to generate a set of random networks. Within-module connectivities, defined as the sum of link weights with each other gene in a gene's module, were calculated for each gene in each random network. The p-value was calculated as the probability of finding within-module connectivity values from this null distribution equal to or greater than the observed within-module connectivity. These p-values were then adjusted using the Benjamini-Hochberg method and genes with adjusted p-values less than 0.05 were designated hub genes.

To compute gene module memberships, eigengenes were first determined for each module using the R package *WGCNA* v1.70–3 (*Langfelder and Horvath, 2008*), and function *moduleEigengenes* then the Pearson correlation was computed between each module eigengene and each gene. Module preservation statistics were calculated using the *WGCNA* function *modulePreservation* (*Langfelder et al., 2011*).

## Gene set enrichment analysis

Gene Ontology lists were obtained from the Molecular Signatures Database (set C5, version 7). Gene set enrichment analysis was performed using the R package *fgsea* v1.18.0 (*Korotkevich et al., 2021*). T-statistics from *voom* regression or gene module membership scores from *MEGENA* were input into the function *fgseaMultilevel* (minSize = 20, maxSize = 200). Resulting pathways were filtered for those with an adjusted p-value less than 0.2 and similar pathways were pruned using *collapsedPathways* (pval.threshold=0.01 or 0.05).

## Inter-region correlation analysis

To analyze inter-region gene correlations, we first selected in each region the 500 genes with the highest variability, computed as the variance-mean ratio of non-log expression across samples. For each gene, we calculated Pearson correlation values for each pairwise combination of brain regions,

yielding region-by-region correlation matrices. To generate null distributions for each gene, we shuffled bird identities for each pairwise region-region comparison 100 times and computed Pearson correlations. We thresholded observed correlations using statistics from these shuffled distributions (correlation lesser or greater than the 2.5% or 97.5% shuffled quantiles). We calculated the across-bird expression similarity between regions as the number of thresholded correlations.

To determine if deafening alters inter-region gene expression coupling in the song system, we computed pairwise Pearson correlations for each gene between each region for hearing and deaf birds separately, then took the absolute value for each matrix. The hearing absolute correlation matrix was then subtracted from the deaf absolute correlation matrix. This procedure was repeated on 100 shuffled distributions to generate a null distribution of differential absolute correlations. Differentially correlated genes were called as those with a deaf versus hearing value less than (decorrelation) or greater than (correlation gain) extreme values of a shuffled distribution calculated for each pairwise comparison (2.5% or 97.5%, respectively).

## Cell type specificity and differential expression scores

For each gene, a specificity score was calculated as $\sum x_n log\left(x_n/\bar{x}_n\right)$, where $x_n$ is expression divided by the sum of expression across all clusters and $\bar{x}_n$ is the mean of this value. Regression coefficient-cell type specificity scores were calculated by selecting differentially expressed genes (adjusted p-value <0.1) and then splitting genes by the sign of the coefficient. Scores were then computed as the dot-product between the cell type × gene specificity matrix and the gene × coefficient matrix.

## Fluorescent in situ hybridization (FISH)

FISH was performed using the hairpin chain reaction system from Molecular Instruments. Birds were euthanized using isoflurane, decapitated, and debrained. Brains were flash-frozen in –70 °C dry ice-chilled isopentane for 12 s within 4 min from decapitation then stored at –80 °C. Fresh-frozen brains were cryosectioned at 16 µm onto SuperFrost slides (Fisherbrand) chilled in the cryochamber then melted onto the slide using a warmed metal dowel. Slides were then transferred to –80 °C for storage. For the FISH, slides were transferred from –80 °C to slide mailers containing cold 4% PFA and incubated for 15 min on ice. Slides were washed three times for 5 min using DEPC-treated PBS + 0.1% Tween-20, dehydrated in 50%, 70%, and two rounds of 100% ethanol for 3–5 min each round, then air dried. Slides were then transferred to a SlideMoat (Boekel Scientific) at 37 °C. 100 µL of v3 Hybridization Buffer (Molecular Instruments) was added to each slide, which were then coverslipped and incubated for 10 min at 37 °C. Meanwhile, 2 nM of each probe was added to 100 µL Hybridization Buffer and denatured at 37 °C. Pre-hybridization buffer was removed, 100 µL of probe/buffer was added, and slides were coverslipped and incubated overnight at 37 °C. The next day, coverslips were floated off in Probe Wash Buffer (PWB, 50% formamide, 5 x SSC, 9 mM citric acid pH 6.0, 0.1% Tween-20, 50 µg/ml heparin), then washed in 75% PWB/25% SSCT (5 x SSC, 0.1% Tween-20), 50% PWB/50% SSCT, 25% PWB/75% SSCT, 100% SSCT for 15 min each at 37 °C. This was followed by 5 min at room temperature in SSCT. Slides were incubated in 200 µL of Amplification Buffer (provided by the company) for 30 min at room temperature. Alexa fluor-conjugated DNA hairpins were denatured for 90 s at 95 °C then allowed to cool for at least 30 min in the dark at room temperature. Hairpins were added to 100 µL amplification buffer, applied to slides, and incubated overnight at room temperature. The following day, slides were washed in SSCT containing 1 ng/mL DAPI for 30 min at room temperature, then SSCT for 30 min at room temperature, followed by a final 5 min in SSCT at room temperature. Prolong Glass Antifade Medium (Thermofisher) was added to each slide and then coverslipped. Sections were imaged on a confocal microscope (Zeiss 710) using a 20 X objective.

## FISH quantification

Image quantification was performed using CellProfiler v4.0.4 (*Stirling et al., 2021*). DAPI-stained nuclei were first identified using the ClassifyPixels-Unet module. Areas corresponding to cells were estimated by extending nuclei boundaries by five pixels. Then signal puncta for each channel were identified and their intensities were measured. For each cell and each channel, we calculated the summed signal intensity of overlapping puncta divided by the cell area. To test for significant differences in gene expression between hearing and deaf birds, a linear mixed effects model was fit using function *lmer* from R package *lmerTest v3.1–3* (*Kuznetsova et al., 2017*) for each target gene and

brain region as 'intensity ~ condition + (1|bird)' where 'condition' is contra or ipsi and '(1|bird)' is the per-bird grouping factor. p-values were obtained by comparing this model with a reduced model 'intensity ~ (1|bird)' using ANOVA.

## Code availability

Code underlying the analysis of birdsong and SLCR-seq gene expression can be found in the GitHub repository https://github.com/bradleycolquitt/deaf_gex (copy archived at *Colquitt, 2023*).

## Acknowledgements

We would like to thank Andrea Hausenstaub and Christoph Schreiner for providing critical commentary on this manuscript, Adria Arteseros for providing technical expertise, and Mimi Kao for surgical expertise.

## Additional information

### Competing interests

Foad Green: Foad Green is affiliated with Syapse, Inc. The author has no financial interests to declare. The other authors declare that no competing interests exist.

### Funding

| Funder | Grant reference number | Author |
|---|---|---|
| National Institute of Neurological Disorders and Stroke | F32NS098809 | Bradley M Colquitt |
| Howard Hughes Medical Institute | Investigator | Michael S Brainard |

The funders had no role in study design, data collection and interpretation, or the decision to submit the work for publication.

### Author contributions

Bradley M Colquitt, Conceptualization, Resources, Data curation, Software, Formal analysis, Validation, Investigation, Visualization, Methodology, Writing – original draft, Project administration, Writing – review and editing; Kelly Li, Resources, Investigation; Foad Green, Robert Veline, Investigation; Michael S Brainard, Conceptualization, Resources, Supervision, Funding acquisition, Project administration, Writing – review and editing

### Author ORCIDs

Bradley M Colquitt http://orcid.org/0000-0001-5819-7924

### Ethics

All Bengalese finches (Lonchura striata domestica) were from our breeding colonies at UCSF or were purchased from approved vendors. All birds experienced a 14 hr:10 hr day:night cycle and were housed in communal cages separated by sex. Experiments were conducted in accordance with NIH and UCSF policies governing animal use and welfare (IACUC protocol number AN107972). All surgery was performed under isoflurane anesthesia, and every effort was made to minimize suffering.

### Decision letter and Author response

Decision letter https://doi.org/10.7554/eLife.85970.sa1
Author response https://doi.org/10.7554/eLife.85970.sa2

## Additional files

### Supplementary files

• Supplementary file 1. SLCR-seq barcode and index sequences. Full sequences for RT_primer_v1,

RT_primer_v2, and PCR2.

• Supplementary file 2. Deafening voom statistics Fold-change estimates and p-values for voom regression analysis of the deafening SLCR-seq dataset.

• Supplementary file 3. GSEA statistics Gene set enrichment analysis of song destabilization-associated genes.

• Supplementary file 4. Network module memberships Gene memberships in MEGENA modules for the RA, HVC, LMAN, and Area X networks.

• Supplementary file 5. Differential correlation analysis Inter-region gene correlations results for the combined dataset and split between hearing and deaf birds.

• Supplementary file 6. Unilateral LMAN lesion voom statistics Fold-change estimates and p-values for voom regression analysis of the unilateral LMAN lesion SLCR-seq dataset.

• MDAR checklist

## Data availability

SLCR-seq mapped sequencing reads, gene-by-sample count matrices, and metadata can be found at NCBI GEO for deafening (accession number GSE200663) and unilateral LMAN lesion datasets (GSE200664).

The following datasets were generated:

| Author(s) | Year | Dataset title | Dataset URL | Database and Identifier |
|---|---|---|---|---|
| Colquitt BM, Brainard MS | 2022 | Analysis of the effects of deafening on gene expression in birdsong neural circuitry | https://www.ncbi.nlm.nih.gov/geo/query/acc.cgi?acc=GSE200663 | NCBI Gene Expression Omnibus, GSE200663 |
| Colquitt BM, Brainard MS | 2022 | Analysis of the effects of unilateral LMAN lesioning on gene expression in birdsong neural circuitry | https://www.ncbi.nlm.nih.gov/geo/query/acc.cgi?acc=GSE200664 | NCBI Gene Expression Omnibus, GSE200664 |

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
