## [Editor Report]

This is an important study that uses the song system in a bird model to understand the transcriptional mechanisms underlying neuronal adaptations to sensory deprivation. The manuscript offers compelling data in support of the authors' hypothesis that these transcriptional changes are related to song plasticity. The work will be of interest to biologists who study neuronal plasticity mechanisms.

---

## [Decision Letter]

**Decision letter after peer review:**

Thank you for submitting your article "Neural circuit-wide analysis of changes to gene expression during deafening-induced birdsong destabilization" for consideration by *eLife*. Your article has been reviewed by 3 peer reviewers, and the evaluation has been overseen by a Reviewing Editor and Andrew King as the Senior Editor. The reviewers have opted to remain anonymous.

Essential revisions:

1) All three reviewers raised questions about how gene expression changes related to aspects of neural circuit function and song production. Please read the comments below and address these points in a revised manuscript.

2) The reviewers offer suggestions for some minor corrections/clarifications for the figures.

*Reviewer #1 (Recommendations for the authors):*

I have no concerns with the data or the analysis, but just one question that the authors may want to consider discussing. Especially in the case of gene expression changes in the interneurons following deafening or in RA following LMAN lesions, could the gene expression changes the authors see primarily result from a decrease in neuronal firing? Expression of some neuropeptides is known to be activity-dependent, and the identification of the activity-dependent gene BDNF as one that shows reduced correlations between RA and LMAN after deafening could also reflect changes in neuronal firing rather than being a plasticity factor that drives adaptations in the circuit. I am curious if the authors can discuss possible ways to disambiguate whether the transcriptional changes are a cause or an effect of the change in song structure.

*Reviewer #2 (Recommendations for the authors):*

This study tackles a significant question, introduces a novel and innovative methodology to do so, and is clearly written and beautifully presented. The findings of the study are compelling and will be of broad interest to scientists studying the genetics of vocal communication, as well as scientists studying the genetics of motor skill learning more generally. I have only a small number of comments/questions for the authors:

Major comments

– What is the rationale for including 3 time points post-deafening, given that time post-deafening isn't explicitly considered in most of the analyses of gene expression changes? Are the deaf and hearing birds that overlap in Dkl values those that were sacrificed at the earliest post-procedure timepoint (and that were subsequently excluded from gene expression analyses)? I am not suggesting that the authors change the design or perform additional experiments, but it would be helpful for the reader to more clearly explain the rationale behind including 3 post-procedure time points.

– Can we rule out the possibility that changes in gene expression are due to song destabilization vs. the complete absence of auditory input? Would the authors expect to observe similar changes in gene expression if an alternate method were used that disrupted song stability but preserved hearing (ts nerve cut, etc.)? Conversely, I'm curious what the authors think they would see if they combined bilateral LMAN lesions with deafening (a combination of manipulations that should block song degradation but will still abolish hearing). Again, I'm not suggesting that the authors would perform these additional experiments, but perhaps an expanded discussion of these possibilities added to the discussion would be of interest to readers.

– What is the significance of the relationship of pre-procedure singing rate on LMAN gene expression (measured 9-14 days later; Supplemental Figure 3-1B)? I'm unclear on why singing rates so far in the past would relate LMAN gene expression measured days/weeks later. This is confusing/interesting, particularly given that LMAN gene expression seems much more weakly related to the pre-euthanasia singing rates.

*Reviewer #3 (Recommendations for the authors):*

The manuscript is extremely well written, the figures are clear and visually compelling, and the existing analysis is reasonable and rigorous. The code is available via GitHub, and methodological details including reagent sources are sufficient. A minor suggestion would be to clarify instances of "described above" (example: line 644) as there is a lot of information here and many things could have been previous. Addressing the outstanding questions outlined in the Public Review would increase the impact of this study, as it would deepen some of the analysis and connect potential mechanisms to long-standing data in the songbird system, and tighten the relationship between the current findings and behavioral outcomes.

---

## [Author Response]

Reviewer #1 (Recommendations for the authors):I have no concerns with the data or the analysis, but just one question that the authors may want to consider discussing. Especially in the case of gene expression changes in the interneurons following deafening or in RA following LMAN lesions, could the gene expression changes the authors see primarily result from a decrease in neuronal firing? Expression of some neuropeptides is known to be activity-dependent, and the identification of the activity-dependent gene BDNF as one that shows reduced correlations between RA and LMAN after deafening could also reflect changes in neuronal firing rather than being a plasticity factor that drives adaptations in the circuit. I am curious if the authors can discuss possible ways to disambiguate whether the transcriptional changes are a cause or an effect of the change in song structure.

The reviewer brings up an excellent point concerning resolving the effects of altered neural activity and motor plasticity on gene expression. We controlled for recent singing-elicited neural activity by including the number of songs sung in the two hours prior to euthanasia in our regression analysis. We found no significant difference between hearing and deafened birds for this value (Figure 3B). This singing-based measure is a useful proxy for neural activity in the song system, however it is true that variation in other activity patterns, e.g. spontaneous activity, could also contribute to the observed gene expression differences. A useful set of future experiments could help resolve this issue by directly altering neural activity in the song system and analyzing its effects on expression. We now describe alternative methods to drive song destabilization (e.g. delayed auditory feedback and tracheosyringeal nerve cut) that would alter neural activity in a different manner than deafening and could be combined with gene expression analysis approaches used in this study (lines 842-850). Furthermore, we discuss using direct manipulations of the expression of candidate genes to test their causal roles in promoting or limiting song plasticity (lines 850-853).

Reviewer #2 (Recommendations for the authors):This study tackles a significant question, introduces a novel and innovative methodology to do so, and is clearly written and beautifully presented. The findings of the study are compelling and will be of broad interest to scientists studying the genetics of vocal communication, as well as scientists studying the genetics of motor skill learning more generally. I have only a small number of comments/questions for the authors:Major comments– What is the rationale for including 3 time points post-deafening, given that time post-deafening isn't explicitly considered in most of the analyses of gene expression changes? Are the deaf and hearing birds that overlap in Dkl values those that were sacrificed at the earliest post-procedure timepoint (and that were subsequently excluded from gene expression analyses)? I am not suggesting that the authors change the design or perform additional experiments, but it would be helpful for the reader to more clearly explain the rationale behind including 3 post-procedure time points.

We thank the reviewer for this point of clarification and have added text in the Results describing our rationale for the 3 time point design (lines 107-108). We selected these timepoints to generate a cohort of animals with a broad range of song destabilization. Generally, the longer a bird was deafened the higher its Song D_KL_ values (now explained in more detail in the revised manuscript, lines 321-325). As to the hearing and deaf birds that had overlapping Song D_KL_, this group had representatives from all three time-point groups and did not show a bias toward early time points.

– Can we rule out the possibility that changes in gene expression are due to song destabilization vs. the complete absence of auditory input? Would the authors expect to observe similar changes in gene expression if an alternate method were used that disrupted song stability but preserved hearing (ts nerve cut, etc.)?

We agree that with the current dataset it may be difficult to separate expression changes due to song-related plasticity versus the loss of auditory input. Motor-specific manipulations such as tracheosyringeal nerve cut would be an excellent complementary approach to resolve this issue. In the revised manuscript, we have noted this caveat and have presented experiments, such as TS cut, as ways to address it (Discussion, lines 842-850).

Conversely, I'm curious what the authors think they would see if they combined bilateral LMAN lesions with deafening (a combination of manipulations that should block song degradation but will still abolish hearing). Again, I'm not suggesting that the authors would perform these additional experiments, but perhaps an expanded discussion of these possibilities added to the discussion would be of interest to readers.

The reviewer raises an important set of followup experiments that assess the extent to which the transcriptional state of the song system tracks with song plasticity state. Coupling LMAN lesions with deafening, a manipulation that largely prevents song degradation, would be a strong approach to identify genes whose expression is closely tied to song destabilization. We now discuss the possibility of this approach to further identify genes linked with song destabilization in the Discussion (lines 939-946).

– What is the significance of the relationship of pre-procedure singing rate on LMAN gene expression (measured 9-14 days later; Supplemental Figure 3-1B)? I'm unclear on why singing rates so far in the past would relate LMAN gene expression measured days/weeks later. This is confusing/interesting, particularly given that LMAN gene expression seems much more weakly related to the pre-euthanasia singing rates.

We agree that this is an interesting result, and also agree that it is a bit puzzling. We included pre-procedure singing rate in the regression to control for gene expression related to a bird’s propensity to sing, which would be distinct from expression induced by recent singing (separately quantified as the amount of singing in the 2 hours prior to euthanasia). We rechecked the results for LMAN and did not find anything which could have generated aberrant results (e.g. extreme outliers or poor quality data). Because we primarily included this variable as a control, we did not delve deeply into this finding, but agree that it could be worth further investigating this finding in follow-up work.

Reviewer #3 (Recommendations for the authors):The manuscript is extremely well written, the figures are clear and visually compelling, and the existing analysis is reasonable and rigorous. The code is available via GitHub, and methodological details including reagent sources are sufficient. A minor suggestion would be to clarify instances of "described above" (example: line 644) as there is a lot of information here and many things could have been previous. Addressing the outstanding questions outlined in the Public Review would increase the impact of this study, as it would deepen some of the analysis and connect potential mechanisms to long-standing data in the songbird system, and tighten the relationship between the current findings and behavioral outcomes.

We thank the reviewer for their review and recommendations. We have replaced “described above” statements with more informative descriptions to the referred sections of the manuscript.